# An evaluation of daily precipitation from a regional atmospheric reanalysis over Australia

Suwash Chandra Acharya[1], Rory Nathan[1], Quan J Wang[1], Chun-Hsu Su[2], Nathan Eizenberg[2]

[1] Department of Infrastructure Engineering, The University of Melbourne, Melbourne, Australia

[2] Bureau of Meteorology, Melbourne, Australia

*Correspondence to*: S.C. Acharya (suwasha@student.unimelb.edu.au)

**Abstract.**

An accurate representation of spatio-temporal characteristics of precipitation fields is fundamental for many hydro-
meteorological analyses but is often limited by the paucity of gauges. Reanalysis models provide systematic methods of representing atmospheric processes to produce datasets of spatio-temporal precipitation estimates. The precipitation from the reanalysis datasets should, however, be evaluated thoroughly before use because it is inferred from physical parameterization. In this paper, we evaluated the precipitation dataset from the Bureau of Meteorology Atmospheric high-resolution Regional Reanalysis for Australia (BARRA) and compared it against (a) gauged point observations, (b) an interpolated gridded dataset
based on gauged point observations (AWAP), and (c) a global reanalysis dataset (ERA-Interim). We utilized a range of evaluation metrics such as continuous metrics (correlation, bias, variability, modified Kling-Gupta efficiency), categorical metrics, and other statistics (wet day frequency, transition probabilities and quantiles) to ascertain the quality of the dataset. BARRA, in comparison with ERA-Interim, shows a better representation of rainfall of larger magnitude at both point and grid scale of 5 km. BARRA also more closely reproduces the distribution of wet days and transition probabilities. The performance
of BARRA varies spatially, with better performance in the temperate zone than in the arid and tropical zones. A point-to-grid evaluation based on correlation, bias and modified Kling-Gupta efficiency (KGE') indicates that ERA-Interim performs on par or better than BARRA. However, on a spatial scale, BARRA outperforms ERA-Interim in terms of KGE' score and the components of the KGE' score. Our evaluation illustrates that BARRA, with richer spatial variations in climatology of daily precipitation, provides an improved representation of precipitation compared with the coarser ERA-Interim. It is a useful
complement to existing precipitation datasets for Australia, especially in sparsely gauged regions.

## 1 Introduction

Availability of accurate precipitation datasets is an essential requirement for the modelling of natural processes, hydro-meteorological analyses and forecasting, monitoring climatic variations and changes (Kirschbaum et al., 2017; Kucera et al.,

2013; Robertson et al., 2013). A comprehensive knowledge of occurrence and distribution of precipitation is however hindered by the sparseness of the gauging network. Variations in the density and coverage of the gauging network make it difficult to capture information on the spatial and temporal variability of rainfall. This is particularly the case in areas covered by deserts, mountains, and oceans and in large areas with low population densities (Salio et al., 2015; Thiemig et al., 2012). This presents a challenge for the Australian continent where the gauges are mostly located along the densely populated coastal regions. The station network is less dense in the central region which represents the more arid part of the continent (Johnson et al., 2016).

The difficulties inherent in existing observation networks have prompted the development of various gridded datasets with consistent spatial and temporal scale.  One such precipitation dataset for Australia is the interpolated precipitation product from the Australian Water Availability Project (AWAP). Because of the uneven gauge network distribution, it is not consistent in terms of accuracy (Jones et al., 2009). Global and regional reanalysis datasets provide another source of precipitation data at a consistent spatial and temporal resolution. Such reanalysis datasets are generated using a numerical weather prediction (NWP) model and a data assimilation scheme to incorporate the available observations thereby providing a consistent method of representation of the atmosphere at a regular interval over larger spatial and temporal domain (Parker, 2016). A range of global reanalysis datasets such as NCEP-CFSR (Saha et al., 2010), ERA-Interim (Dee et al., 2011), JRA-55 (Kobayashi et al., 2015) are readily available.

Global reanalysis datasets have been evaluated for various applications at global and regional scales. Of the global dataset exclusively based on reanalysis, ERA-Interim is generally considered to provide better performance compared to other reanalysis datasets (Beck et al., 2017, 2019). ERA-Interim has been found to reproduce the climatology of global monsoon precipitation (Lin et al., 2014) and in general it demonstrates high temporal and spatial correlations with interpolated observations (Donat et al., 2014). In a recent global evaluation of gridded precipitation datasets using gauge observations by Beck et al (2017), the ERA-Interim and JRA-55 reanalysis datasets were found to reproduce long-term trends and temporal correlation more reliably than achieved by satellite datasets. In the Australian continent, ERA-Interim reproduced the observed spatial patterns of long-term rainfall along with other climatic variables and showed an overall better performance compared to NCEP-NCAR (National Centers for Environmental Prediction/National Center for Atmospheric Research) reanalysis (Fu et al., 2016). An evaluation by Peña-Arancibia et al. (2013) of reanalysis datasets, satellite products, and an ensemble of these datasets in Australian and Asian regions showed that the ERA-Interim performed better than other individual datasets across a range of metrics for the Australian region.

The available global reanalysis datasets such as NCEP-CFSR, ERA-Interim, and JRA-55 cover the Australian region, but their horizontal resolutions are relatively coarse ($\geq$ 80 km) and unsuitable for fine-scale application in hydro-meteorological analysis. The resolution of the global reanalysis can be enhanced by downscaling approaches such as dynamic downscaling (Soares et al., 2012) or statistical analysis using high-resolution surface observations (Vidal et al., 2010). Alternatively, the

application of a regional model-based data assimilation can provide a better representation of local climate features and extreme events (Bollmeyer et al., 2015). The regional reanalysis, unlike global reanalysis, allows the integration of abundant local surface observations at a finer scale (Bollmeyer et al., 2015; Isotta et al., 2015; Jakob et al., 2017). The studies have also been conducted to evaluate the additional benefit of high-resolution datasets obtained from regional reanalysis. Jermey and Renshaw

(2016) found that the overall performance of 12km regional reanalysis was significantly better than 80km ERA-Interim especially for high thresholds of rainfall. Similarly, the performance of a regional analysis against ERA-Interim at grid scale showed an improvement in representing high-threshold events (Isotta et al., 2015; Jermey and Renshaw, 2016; Roberts and Lean, 2008) and convective events (Roberts and Lean, 2008). In general, regional reanalyses based on boundary conditions from ERA-Interim is shown to improve the performance over ERA-Interim albeit these studies are largely focused in Europe.

The Bureau of Meteorology Atmospheric high-resolution Regional Reanalysis for Australia (BARRA) includes a 12km modelling and assimilation system (BARRA-R; in this paper, we refer to BARRA-R as BARRA for convenience) which is the first 12 km regional reanalysis conducted over Australia, New Zealand and southeast Asia (Su et al., 2019). BARRA is expected to provide an improved understanding of the past weather than previously possible, particularly for extreme events, and this should support better planning and management of climate-related risks in the future. BARRA makes use of local

surface observations and locally derived wind vectors which are not available to global reanalysis models. However, precipitation observations are not assimilated in BARRA. Precipitation is modelled using a microphysics parameterization and a mass flux convection scheme. The modelled precipitation can be erroneous (Bukovsky and Karoly, 2007; Parker, 2016). Therefore, it is necessary to understand the nature of the uncertainty and inaccuracies involved and to use this understanding to correct for any systematic errors.

The criteria for evaluation of the reanalysis precipitation need to be rigorous and relevant to the intended purpose of use (Parker, 2016). The type of reference datasets (point or grid), the spatial and temporal scale, and the metrics used vary widely among the studies. In general, gridded precipitation products including reanalysis datasets, satellite precipitation estimates, and/or interpolated datasets are evaluated against benchmark datasets (Baez-Villanueva et al., 2018; Beck et al., 2017; Gebremichael, 2010; Isotta et al., 2015; de Leeuw et al., 2015; Peña-Arancibia et al., 2013; Zambrano-Bigiarini et al., 2017).

Suitable benchmark datasets comprise point measurements (Baez-Villanueva et al., 2018; Chiaravalloti et al., 2018; Salio et al., 2015; Thiemig et al., 2012; Zambrano-Bigiarini et al., 2017) and/or high-quality gridded datasets (Chiaravalloti et al., 2018; Isotta et al., 2015; Peña-Arancibia et al., 2013). Catchment-scale representativeness of precipitation datasets is assessed by evaluating catchment average precipitation (e.g. Thiemig et al., 2012). Evaluation studies for hydrological applications focus on a daily timescale analysis (Baez-Villanueva et al., 2018; Thiemig et al., 2012; Zambrano-Bigiarini et al., 2017),

though this has been extended to a sub-daily time interval for a purpose of hydrologic risk assessment (Chiaravalloti et al., 2018).

Given the complex statistical behaviour of precipitation, a range of evaluation metrics are available which have their own assumptions and limitations. Typical metrics include unconditional scores (e.g. correlation, bias, root mean square error, mean absolute error), categorical metrics (e.g. probability of detection, false alarm ratio, skill scores) and distributional statistics (Gebremichael, 2010). In addition, statistical properties like quantiles, wet day frequency, and transition probabilities help to

ascertain if the dataset under evaluation preserves the statistical properties related to sequencing. Any single metric cannot adequately represent the nature of all errors in the precipitation products. Therefore, it is essential to evaluate multiple metrics which describe different aspects of precipitation to identify the possible sources of mismatch between datasets (Baez-Villanueva et al., 2018).

This study is one of the first comprehensive explorations of the BARRA precipitation estimates. It aims to identify strengths

and limitations of the BARRA precipitation to ascertain its efficacy for further hydrometeorological applications in the Australian region. The evaluation is performed against gauged measurements from two data sources: one is based on gauged point observations, and the other is a high-resolution gridded AWAP dataset derived from interpolating gauge measurements, which is widely accepted as being the best synthesis of gauged observations. There are inherent differences in gridded and point rainfall estimates due to the spatial averaging of point observations across each grid cell area. Since BARRA provides

direct estimates of area-average rainfall, it is useful to compare these estimates with the best available point and areal precipitation estimates. In addition to observed and interpolated datasets, we also compare BARRA with corresponding areal-rainfalls provided by the ERA-Interim global reanalysis. ERA-Interim was selected as it is used to drive BARRA (Su et al., 2019) and thus this comparison reveals the extent to which the high-resolution regional reanalysis provides additional value over ERA-Interim in capturing finer-scale meteorology. ERA-Interim was also selected as it has been found to be one of the

best performing reanalysis datasets (Peña-Arancibia et al., 2013).

Despite the availability of BARRA precipitation as hourly values, we select a daily timescale to match the temporal resolution of the best available gridded reference dataset (AWAP). In addition, the accuracy at a daily scale serves as a desirable first step towards further examination at finer timescales because any reliance on sub-daily estimates necessarily depends on its ability to correctly represent daily precipitation. The additional value of evaluation at a daily scale is that it assesses the

potential of BARRA to provide estimates of daily rainfall in the sparsely gauged regions across Australia. We focus our evaluation on a range of metrics related to the suitability of rainfalls for hydrological applications and climate studies. We consider the depth, statistical distribution, wet day frequency and transition probabilities at a daily level for evaluation. The purpose of this evaluation is thus to shed light on the suitability of BARRA for input to hydro-meteorological applications related to climate change studies, water resource management, and the analyses of floods and droughts.

## 2 Datasets

The sub-hourly time series of recorded rainfall from continuous rainfall stations are obtained from the Bureau of Meteorology. Daily rainfall is generated by aggregating the sub-daily rainfall observations. The period of analysis was determined by the duration at which all the datasets used in the study are available i.e. January 2010 to December 2015. The gauge stations used in the study are chosen based on the availability of information for the entire period over which BARRA data estimates are available. A total of 441 stations are selected, and their spatial distribution is shown in Figure 1(a).

An overview of gridded datasets used in this study is presented in the Table 1.The Australian Water Availability Project (AWAP) provides a daily high-quality 0.05 ° × 0.05 ° (around 5 km) gridded rainfall product, dating back to 1900, based on an extensive network of rain gauges, as described by (Jones et al., 2009). The AWAP product makes use of daily (9 am to 9 am) data from all available gauges across the whole of Australia. The gridded estimates are obtained using a weighting scheme that incorporates an optimized Barnes successive-correction algorithm and orographic corrections (Jones et al., 2009). AWAP uses land-based observations only and therefore does not provide information over the ocean.

The global reanalysis ERA-Interim of the European Centre for Medium-Range Weather Forecasts (ECMWF, www.ecmwf.int) covers the period from 1 January 1979 to present (Dee et al., 2011). The core component of the ERA-Interim data assimilation system is the 12h 4D-variational (4DVar) analysis scheme of the upper-air atmospheric state, which is on a spectral grid with triangular truncation of 255 waves (corresponding to approximately 80 km) and a hybrid vertical co-ordinate system with 60 vertical levels. The precipitation is estimated by the numerical model based on temperature and humidity information derived from assimilated observations originating from Passive Microwave and in-situ measurements. In this study, the sub-daily precipitation from ERA-Interim is converted to daily by accumulation to 24 hours to be consistent with the gauged and AWAP data.

BARRA also uses 4DVar but at the 36 km resolution, and uses the Unified Model as its forecast model (Su et al., 2019). BARRA extends spatially over 65.0° to 196.9° east, -65.0° to 19.4° north at a spatial resolution of 0.11° (approximately 12 km) and with 70 levels up to 80 km into the atmosphere. The project aims to produce the dataset dating back to 1990, however, at present, only a six-year period (2010-2015) is available. The model includes a comprehensive set of parametrisations, including a modified boundary layer scheme, mixed phase cloud microphysics, a mass flux convection scheme, and a radiation scheme. The model parametrisation in BARRA mainly is inherited from the UKMO Global Atmosphere (GA) 6.0 configurations as described in Walters et al. (2017). Surface and satellite rainfall observations are not assimilated, and the precipitation fields are determined using the assimilated large-scale variables and the physical parameterisation of the model. At 12 km horizontal resolution, BARRA requires the convection scheme to model sub-grid scale convection using an ensemble of cumulus clouds as a single entraining-detraining plume (Clark et al., 2016). The scheme prevents unstable growth of cloud

structures on the grid and explicit vertical circulations and can only predict an area-average rainfall instead of a spectrum of rainfall rates. The analysis for BARRA is conducted 4 times a day with a 6-hour analysis window centred at time $t_0 = 0, 6, 12$ and 18 UTC. The forecast cycle is of 12 hours which ranges from $t_0$-3h to $t_0$+9h (Su et al., 2019). The BARRA precipitation forecasts from $t_0$+4 hour to $t_0$+9 hour are used to generate a daily time series for the study.

## 3 Methodology

### 3.1 Reference Datasets

The study is conducted over a period of 6 years, from January 2010 to December 2015, which represents the full period of available data at the commencement of the study. We undertake a point-to-grid and grid-to-grid evaluation of the reanalysis datasets against suitable reference datasets. The gauged data represents the best estimate of rainfall at a point, and AWAP data provides the best estimate of rainfall over a grid cell. Both estimates are, however, limited by the available gauging density. AWAP estimates are based on all available daily and continuous gauges, and the point data considered here are from selected sub-daily rainfall stations with maximum availability of data over the study period. The selected gauges are distributed unevenly across Australia and are spatially representative of the available gauging network. As the gridded datasets represent an areal average, it may be expected that there are differences between point and gridded estimates as the latter account for some spatial averaging. While the gauged data and AWAP rainfall estimates represent the best available reference datasets based on measured data, both are imperfect representations of areal rainfalls. The AWAP estimates contain inaccuracies due to the interpolation method, and the point estimates provide only a coarse estimate of rainfall over a grid cell area. The ability of these point and gridded reference data sets to represent actual areal rainfalls is heavily dependent on the gauging density and local orography, and these factors influence the accuracy of the reference data sets to different degrees across Australia. Accordingly, we compare the BARRA and ERA-Interim estimates to both point gauged and AWAP areal data, where the evaluation of both offer different insights to the quality of model estimates.

Gridded rainfalls are compared to point rainfalls using a nearest neighbour approach. The choice of interpolation scheme is especially important when comparing datasets of different spatial resolutions; bilinear interpolation is likely to smoothen the precipitation field resulting in higher bias for larger rainfall (Accadia et al., 2003), whereas nearest neighbour method preserves the magnitude of the precipitation over the grid. At each location of point rainfall, the nearest neighbour method assigns the precipitation amount from the nearest grid point. In grid-to-grid evaluation, both reanalysis datasets are interpolated to a common AWAP spatial scale using nearest neighbour method. That means, for each AWAP grid, the precipitation variable is obtained from the nearest grid of reanalysis datasets. Given the spatial inconsistency in the accuracy of AWAP dataset, the grid-to-grid evaluation is limited to grid points that are nearest to the gauge stations used in this study.

The performance of daily precipitation from the reanalysis data is assessed considering all days of the year and by seasons (Summer: DJF, Autumn: MAM, Winter: JJA, and Spring: SON). In addition, evaluation is stratified across three broad climatic zones (arid, tropical and temperate) as defined by the Köppen-Geiger classification (Peel et al., 2007). Most comparisons are undertaken using estimates from every day in the 6-year period, and thus the number of wet days varies with the different datasets and locations considered. However, it is recognized that reanalysis products tend to produce a high number of days with light drizzle, and therefore they over-estimate the frequency of wet days. Thus, following Baez-Villanueva et al. (2018), Ebert et al. (2007) and Zambrano-Bigiarini et al. (2017), a threshold of 1 mm/day is used to classify a day as 'wet' or 'dry'.

### 3.2 Performance indices

The evaluation of precipitation data generally involves an assessment of detection capabilities and biases in the form of continuous and categorical metrics. The continuous metrics used in this study are Modified Kling-Gupta efficiency (KGE') (Gupta et al., 2009; Kling et al., 2012) along with its three individual components: correlation (r), bias ratio (β), and variability ratio (γ). Baez-Villanueva et al. (2018) and Zambrano-Bigiarini et al. (2017) also suggested the use of KGE' and its components for evaluating precipitation datasets as it provides an overall assessment along with an error in the representation of magnitude and variability of the reference precipitation.

The four categorical indices adopted are the probability of detection (POD) or hit rate, false alarm ratio (FAR), critical success (or threat) index (CSI), and frequency bias (fBias). These are evaluated over five different rainfall intensity classes, namely: no rain (<1 mm), light rain (≥1 mm and <5 mm), moderate rain (≥5 mm and <20 mm), heavy rain (≥20 mm and <40 mm) and violent rain (≥40 mm) (Baez-Villanueva et al., 2018; Zambrano-Bigiarini et al., 2017). These standard continuous and categorical metrics are described in Table S2 (in the supplement).

### 3.3 Precipitation statistics

In this study, we evaluate the capacity of the reanalysis datasets to reproduce a range of precipitation statistics related to the frequency of wet days, transition probabilities between wet and dry days, and the distribution of rainfall amounts based on the 90%, 95%, and 99% daily exceedance values.

The transition probabilities considered here are $p_{01}$ and $p_{11}$, which denote the probabilities of a dry day followed by a wet day, and a wet day followed by a wet day, respectively. Consider $X_0$ and $X_1$ are two consecutive days, and $i, j$ are two states (0: dry, 1: wet), then the transition probability is expressed as

$$p_{ij} = Pr(X_1 = j | X_0 = i)$$

For dry-wet day: $p_{01} = Pr(X_1 = 1 | X_0 = 0)$

For wet-wet day: $p_{11} = Pr(X_1 = 1 | X_0 = 1)$

## 4 Results

### 4.1 Mean precipitation

Figure 1 shows the average annual precipitation over Australia for the period of six years (2010-2015) as estimated using the
different data sets (gauged point rainfalls, AWAP, BARRA, and ERA-Interim). It is seen that there is a high spatial variability
in the average precipitation over Australia. Regions of high rainfall (northern and eastern coasts along with western Tasmania)
are similarly represented by all four datasets. ERA-Interim provides a coarser representation of the precipitation field, which
expectedly fails to capture the higher spatial variability in the coastal regions and orographic precipitations in the Great
Dividing ranges and western Tasmania. By comparison, BARRA precipitation captures this variability in the AWAP
precipitation. It should be noted, however, that the AWAP data provides a poor estimate of precipitation over central Australia
where there is a paucity of gauging information. BARRA, on the other hand, provides high-resolution precipitation pattern
over the ocean as well as the central Australian region where gauges are sparse.

### 4.2 Point-to-grid assessment

*KGE' and its components*

The variation of KGE' at the gauge locations is shown in Figure 2. As expected, the higher performance of KGE' is observed
in the AWAP dataset as this is derived by interpolating gauged data, which includes the selected high-quality point rainfall
gauges. The pattern of KGE' score looks similar among both reanalysis datasets. The performance, however, varies spatially
with a better score for the gauges located in the southern region in comparison to the central and northern region. A difference
in KGE' score between BARRA and ERA-Interim (Figure 2d) shows the difference between the performance of reanalysis
datasets in various regions. In most of the locations, the difference is minimal indicating the similar performance by both
reanalysis datasets. ERA-Interim generally performs better in the central arid region, whereas BARRA exhibits better scores
in the temperate region. The central arid region experiences very few rainy days which is likely to be missed spatially by a
sparse network of gauges. This could result in poor correlation metrics as it is sensitive to outliers, and subsequently yields a
poor KGE' score. BARRA precipitation is more erroneous in the tropics than in temperate regions which may reflect the
limitations of the convection parameterisation adopted in BARRA. This spatially varying performance of BARRA is further
discussed in section 5.5).

Figure 3 presents the summary of KGE' and its components at a daily scale calculated with reference to gauge data for overall,
summer, and winter seasons. In using all the data at a daily scale, the AWAP precipitation proves to be the best estimate at
gauge location with the highest value of KGE'. In decomposing the overall performance of KGE' into its components (linear

correlation (r), bias ratio (β), and variability ratio (γ)), AWAP achieves the best score for the correlation and variability terms. The AWAP estimates of point rainfall are higher than the gauged observations, and a similar degree of overestimation is exhibited by the BARRA dataset. The Barnes interpolation scheme used in AWAP slightly inflates the spatial coverage of light intensity rainfall (Jones et al., 2009; King et al., 2013). In addition, the weight function assigned to gauged point rainfalls

may result in wetter bias in the densely gauged (coastal) regions compared to other regions. Despite having a slightly lower correlation compared to ERA-Interim, the variability of the rainfall is better captured by the BARRA dataset. The overall KGE' score is slightly better for ERA-Interim compared to BARRA dataset.

Evaluation of daily rainfall in different seasons reveals a mixed performance between the different reanalysis products. The performance during winter is better than in summer for both reanalysis datasets. This is likely due to the ability of NWP models

to accurately simulate synoptic systems which represent the majority of wintertime rainfall. However, this difference is larger for BARRA than ERA-Interim. There is less discrepancy in the correlation between BARRA and ERA-Interim. However, the variation of correlation across stations is higher for the BARRA dataset. BARRA tends to overestimate the depth of rainfall and the ERA-Interim under-estimates variability component across all the seasons. The mean and variability for BARRA during winter (JJA) are closer to gauge estimates than in summer (DJF) season. Based on KGE' score, the performance of

BARRA is lower than ERA-Interim during summer whereas similar during the winter.

The summary of the KGE' scores based on the climatic zone is presented in Table 2. As expected, the KGE' scores are better for AWAP than for BARRA and ERA-Interim for all three climatic zones. The reanalyses match AWAP most closely for the Temperate zone, followed by the Arid zone and performance is worst for the Tropical zone. The KGE' metric shows that BARRA performs better at the temperate zone while ERA-Interim at the tropical zone.

*Wet day frequency and transition probabilities*

Figure 4 shows the comparison of the wet day frequency and transition probabilities ($p_{01}$, dry day followed by a wet day and $p_{11}$, wet day followed by a wet day) at different gauge locations. Overall, BARRA shows a better correlation to the gauge estimates than ERA-Interim and is less biased except for dry-wet transition probability. As expected, the frequency of wet days and transition probabilities recorded at gauge locations are lower compared to all the gridded datasets. Wet day frequency

in all gridded dataset exhibit positive bias with coarser dataset showing higher bias. The difference is higher for the gauge stations in the northern region. From the colour of scatter, it can be observed that the frequency of wet days is lower over central Australia and higher over the south-east coast and Tasmania. The level of agreement between ERA-Interim and gauge estimates of wet day frequencies varies with the proportion of wet days. At the gauges where the frequency of the wet days is very low (<0.2), the estimates from the ERA-Interim are closer to the gauge. However, for the remaining gauges with higher

wet day frequencies, the estimation is positively biased and highly scattered. The wet day frequency varies spatially with the

location of the station. The spatial pattern, however, is similar for all datasets. Transition probabilities are also dependent on the location of stations with a similar overall trend for all datasets. The model estimates of transition probability $p_{11}$ are greater than $p_{01}$. Due to the tendency of NWP models to yield more frequent persistent light rainfalls, a high wet-wet frequency is observed. The variation of $p_{01}$ and $p_{11}$ in the northern region is greater than the southern region. BARRA and AWAP both exhibit higher $p_{01}$ values than gauges for all locations. The $p_{01}$ for the northern region is estimated correctly by ERA-Interim, however, it exhibits more bias and spread for the southern region. In comparison to $p_{01}$, $p_{11}$ shows more scatter and bias. The general trend of overestimation of $p_{11}$ is similar across all datasets and the estimates are higher for the gauges located in northern region compared to southern. The estimates of $p_{11}$ are highly correlated for AWAP. The spatial pattern and correlation are similar for both reanalysis datasets while BARRA exhibits less bias than ERA-Interim.

*Quantiles*

Figure 5 shows the comparison of quantiles obtained from the gridded datasets to the corresponding quantiles derived for the observed point rainfalls. Since all days were considered in computing the quantile (i.e. both wet and dry days were considered), the 99% value represents "large" rainfalls that are exceeded on average only three or four times per year. On the other hand, the rainfall represented by the 90% quantile (exceeded on average 36 times per year) corresponds to the more frequent wet day rainfalls which depend upon the climatology of the station considered. The AWAP rainfall corresponding to these quantiles are all higher than the point rainfall estimates. The quantile estimates in BARRA and ERA-Interim shows very different patterns to those of the AWAP data set. BARRA estimates for all quantiles (90%, 95%, and 99%) exhibit no consistent bias with location. In contrast, ERA-Interim greatly underestimates the larger precipitation, where the degree of underestimation increases with increasing quantile. The nature of the differences varies with location and quantile: for example, the ERA-Interim 90% quantile estimates are biased low compared to the point rainfalls in the northern region, yet there is a tendency for all higher quantiles to be biased high. The ERA-Interim reanalysis appears unable to represent higher precipitation magnitudes, and the largest value plateaus at about 40mm for the rarer quantiles.

*Categorical Metrics*

Figure 6 shows the boxplot of four categorical performance indices computed between three gridded datasets with reference to gauged point rainfalls at a daily scale. The probability of detection (POD) is highest for the AWAP dataset, and the POD values for the two reanalysis data sets are consistently lower and somewhat similar. For rainfall intensities greater than 20 mm and lower than 5mm, BARRA shows higher detection of rainfalls, whereas, for the rainfall intensity 5-20 mm, ERA-Interim shows slightly better detection skill. In contrast, the false alarm ratio (FAR) is higher for BARRA for the heavy rainfall classes (> 20 mm). For the low rainfall classes (<20 mm), FAR is lower for BARRA. The Critical Success Index (CSI) shows that the reanalysis data are not able to adequately represent the distribution of rainfalls greater than 1 mm/day. AWAP, as expected,

has higher CSI for each of the rainfall classes. BARRA has higher CSI for all rainfall classes than ERA-Interim except the moderate rainfall (5-20 mm/day) where both reanalysis datasets have similar CSI. The frequency bias shows the positive bias for AWAP and BARRA for all the rain days. ERA-Interim shows a negative bias for higher rainfall intensities (>20 mm).

### 4.3 Grid-to-grid analysis

*KGE' and its components*

The boxplot of KGE' and its components shown in Figure 7 is computed between reanalysis datasets with reference to AWAP dataset at a daily scale. Among the components, the correlation is similar for both reanalysis datasets, though BARRA exhibits a slightly wider range of correlations than ERA-Interim. However, the bias ratio ($\beta$) and variability ratio ($\gamma$) obtained from BARRA is closer to 1, which is appreciably better than the corresponding statistics for ERA-Interim. The difference in the

overall KGE' measure for the two data sets is not as pronounced compared to its components.

In Table 2, a summary of the KGE' score is also presented with reference to the AWAP dataset. The reanalysis datasets show mixed performance depending on the climate zone and the metrics used. BARRA shows improved performance over ERA-Interim in the temperate zone with better correlation, bias ratio, variability ratio, and KGE' statistics. The difference in correlation coefficients between the reanalysis datasets is higher for the tropical zone but similar for the arid and temperate

zones. An assessment of correlation at grid scale is presented in Figure S1 (in the supplement). It also shows a very high correlation in the southern region and a low correlation at the northern region, although a similar spatial pattern is observed for both reanalysis datasets. Further, the BARRA data set exhibits smaller bias than ERA-Interim across all climatic zones. The variability ratio, however, shows contrasting patterns for the two reanalysis datasets. ERA-Interim shows a marked underestimation in variability across all zones. On the other hand, BARRA closely represents the variability in arid and

temperate zones with overestimation in the tropics.

*Wet day frequency and transition probabilities*

The frequency of wet day and transition probabilities obtained from reanalysis datasets at AWAP grid locations are shown in Figure 8. Overall, the representation of wet day frequency and transition probabilities follow a similar pattern to the point-to-grid assessment (Figure 4). In comparing the gridded datasets with AWAP data as the benchmark, the estimates of wet day

frequency and transition probabilities from BARRA are closer than those of ERA-Interim. However, both reanalysis datasets show an improved correlation and reduced bias in grid-to-grid over point-to-grid evaluation. This is expected as this bias is attributed to the spatial resolution of the data.

*Quantiles*

The comparison of quantiles in gridded datasets in Figure 9 shows a similar pattern as observed in Figure 5 for the gauged point rainfalls. Both reanalysis datasets represent the 90% quantile reasonably well. However, the difference between the reanalysis products increases at higher quantiles. The degree of bias in ERA-Interim is considerably more pronounced compared to BARRA.

5 *Categorical Metrics*

Figure 10 shows the boxplot of four categorical performance indices computed between reanalysis datasets with reference to AWAP dataset at a daily scale. In general, the POD for both datasets is similar to point-to-grid comparison. For the larger precipitation intensities, the detection capacity of BARRA is higher whereas, for the smaller rainfall intensities, the detection from both datasets is close to each other. The FAR scores for BARRA estimates of higher rainfalls are slightly greater, indicating the larger occurrence of false alarms while detecting higher rainfall. For the low rainfall classes, BARRA exhibits slightly better FAR than ERA-Interim. The CSI score is higher for BARRA across all rainfall classes and indicates that BARRA is slightly more skilful than ERA-Interim. At larger rainfall class, the CSI is notably higher for BARRA indicating its better performance in representing larger rainfall. The frequency bias shows that the BARRA precipitation tends to produce more events of light rainfall while missing out on some larger rainfalls. The bias in BARRA is, however, smaller in comparison with ERA-Interim which shows a marked underestimation of larger rainfall events (>20 mm).

## 5 Discussion

The evaluation of daily precipitation from BARRA and its comparison against existing datasets reveals a mixed performance under varying benchmark dataset and the metrics. The general observations on the performance of BARRA are summarised in Table 3, and the key insights from these results are discussed below.

### 5.1 Representation of spatial precipitation structure

The BARRA dataset exhibits similar spatial patterns of mean annual rainfall as the AWAP dataset in regions where gauge density is highest (Figure 1). However, the reliability in the evaluation of BARRA in the central regions of Australia is confounded by the lack of gauges. The coarser resolution of ERA-Interim misses the small-scale variability; in contrast, the more realistic representation of complex topography in BARRA helps to capture fine-scale features such as orographic precipitation (Su et al., 2019). A good level of agreement in spatial patterns of rainfalls with AWAP and similarity with ERA-Interim estimates (over both land and ocean) at a coarser scale suggests that the BARRA dataset might provide useful information on the distribution of rainfall in regions where direct measurement of precipitation is not available.

## 5.2 Representation of average precipitation and variability

In general, a slight overestimation is observed in mean rainfall from high-resolution gridded data when compared to gauged point rainfall estimates (bias in mm/day for AWAP = 0.48, BARRA = 0.43). In AWAP, the wet bias observed at low rainfall is due to the Barnes interpolation analysis (Jones et al., 2009). In addition, the weight functions tend to spread rainfalls across the grid which tends to induce a wet bias in coastal stations since the density of stations near the coast is greater. When average AWAP precipitation is considered a benchmark for evaluation, BARRA (R=0.93, bias = -0.05mm/day) shows better agreement than ERA-Interim (R=0.84, bias = -0.49 mm/day). In the locations with higher average precipitation, the ERA-Interim underestimates gauged point rainfalls. This can be attributed to the coarser resolution of ERA-Interim; an 80km scale areal averages are expected to be less than observed for rainfall systems that are spatially localized and/or heterogeneous.

In addition, the temporal variability is represented well by BARRA while this is underestimated by ERA-Interim (Figure 3 and Figure 7). As with the assessment of spatial variability, the reason for this underestimation is the coarser resolution of ERA-Interim. The variability ratio varies across climatic zones with largest difference in the tropical zone. BARRA greatly overestimates variability in the tropical region, whereas the opposite is true for ERA-Interim (Table 2). The discrepancy in the performance across climatic zones is further discussed in section 5.5.

## 5.3 Representation of wet day frequency and transition probabilities

All the gridded datasets exhibit higher frequencies of wet day occurrence compared to gauge point rainfalls. This difference is consistent with the physical reasoning that the likelihood of rainfall occurring over an area is always higher than that over a point location. Moreover, an increase in the area produces a higher difference in the wet days and this is consistent with the increasing levels of bias with grid cell size between AWAP, BARRA, and ERA-Interim (Figure 4). In comparison with AWAP as a benchmark, BARRA closely represents the wet day frequency and exhibits less uncertainty at higher estimates compared to ERA-Interim (Figure 8).

The variation of transition probabilities $p_{01}$ and $p_{11}$ at a location can be attributed to the seasonal distribution of precipitation at that location. Most rainy days in the northern region occur during October-March with very low or no rainfall for the rest of the season. Due to distinct dry (and wet) season, the likelihood of a dry day following a dry day (and a wet day following a wet day) is higher. This results in larger $p_{11}$ and smaller $p_{01}$ estimates. In contrast, there is not a distinct wet and dry season in the south, and $p_{01}$ is close to $p_{11}$ for the southern region, mostly due to the light rainfall distributed over the longer period. The ability of the reanalysis datasets to represent transition probabilities also varies with location due to the climatology and varying precipitation mechanisms across locations. The spread in transition probabilities $p_{01}$ is estimated well by all gridded datasets, whereas the $p_{11}$ estimates are overestimated and more varied. As mentioned above, this difference is due to the tendency of NWP models to over-estimate the persistence of light rainfalls (Kendon et al., 2012). Due to the likelihood of higher frequency

of wet days in gridded datasets, the transition probability $p_{11}$ is higher in gridded datasets and exhibits higher variability for the tropical zone.

## 5.4 Over/Under-estimation of large rainfall events

The comparison of quantiles (Figures 5 and 9) demonstrates that the BARRA dataset is able to represent the higher quantiles
more accurately than ERA-Interim. The BARRA estimates of quantiles show less bias, but variance increases with the magnitude of precipitation amount. This uncertainty could be due to displacement error as the precipitation becomes more localized with an increase in magnitude. ERA-Interim, on the other hand, greatly underestimates the higher quantiles. This could be attributed to its coarser resolution resulting in the averaging of rainfall over a larger area. These results are consistent with published findings (Isotta et al., 2015; Jermey and Renshaw, 2016) that high-resolution regional reanalysis improves over
ERA-Interim in the representation of large rainfall.

The categorical evaluation (Figure 6 and Figure 10) also illustrates the improved performance of the BARRA dataset over ERA-Interim during larger rainfall events. Because of the higher spatial resolution, BARRA provides a more accurate representation of larger rainfalls. Therefore, the probability of detection is higher for the larger rainfall classes. Despite exhibiting a higher hit rate, BARRA also shows higher false alarm ratios. The increased false alarm in BARRA is due to the
higher number of large rainfall events reported by the BARRA dataset.  With larger rainfall events, the false alarm is also likely to increase. Such a trend is usually observed in the assessment of reanalysis and satellite rainfall estimates (Zambrano-Bigiarini et al., 2017). Critical Success Index, the metric which penalizes both misses and false alarms, represents the overall skill of the data. The performance of BARRA lies between the skilful AWAP and less skilful ERA-Interim. The greater CSI for BARRA compared to ERA-Interim suggests that its improved hit rate outweighs the frequency of false alarms. The
difference in CSI between BARRA and ERA-Interim is greater for larger rainfall classes, with the former yielding a better score.

## 5.5 Superior performance over temperate than over tropical and arid

Most metrics suggest that the performance of BARRA is superior in the southern (temperate) region compared to the northern (tropical) region. The summary of KGE' and its component across climatic zones (Table 2) and correlation statistics (Figure S1)
clearly shows the variation in performance over different climatic zones. The possible explanation for this may be the difference in the climatic systems driving the precipitation in those regions and the scheme used to generate precipitation (Su et al., 2019). Convective precipitation is dominant in tropical regions and the parameterisation scheme for sub-grid convection adopted in BARRA is limited in terms of resolving such precipitation. Therefore, a higher accuracy of BARRA can be expected at high latitudes where synoptic rainfall dominates than in low latitudes where convective rainfall is dominant (Ebert et al.,
2007; Su et al., 2019). The limitation of NWP models to represent convective precipitation accurately has also been reported

by Ebert et al. (2007) and de Leeuw et al. (2015). In addition, the network of surface observations used to develop BARRA is denser in the southern region compared to the northern, and this difference in information content also contributes to discrepancies in performance between these regions.

**5.6 Varying performance in point-to-grid and grid-to-grid evaluation:**

This study shows that the general pattern of performance between the reanalysis datasets is similar in the point-to-grid evaluation. However, the evaluation against the gridded AWAP estimates showed a markedly better performance by BARRA in terms of overall bias, variability, wet day frequencies, transition probabilities, quantiles and categorical metrics. As expected, BARRA better represents areal rather than point rainfall as it represents the average precipitation field over a grid cell.

**6 Conclusions**

The purpose of the current study is to document the performance of the BARRA dataset at a daily scale and to provide a comparative analysis of its strengths and limitations relative to other available datasets. The analysis includes point-to-grid and grid-to-grid evaluations at the gauge locations. A range of metrics representing correlation, daily precipitation statistics, and categorical performance are explored and compared on an annual as well as a seasonal basis.

The high-resolution nature of the BARRA dataset provides more detailed and accurate estimates of rainfall across the
Australian region than the coarser ERA-Interim.  BARRA precipitation exhibits good agreement with the average annual rainfall from AWAP as well as to the gauge dataset. The correlation statistics of the BARRA estimates are slightly lower than for the global reanalysis (ERA-Interim). However, the depth and variability of daily precipitation from AWAP are better reproduced by BARRA than by ERA-Interim. We can conclude that BARRA precipitation is representative at the spatial scale of AWAP considering that AWAP data provides the best basis for comparison with the reanalysis datasets.

BARRA provides largely unbiased estimates of larger rainfall quantiles whereas ERA-Interim are clearly underestimated. Categorical evaluation also shows a better correspondence of the larger events in BARRA compared to ERA-Interim. This has important implications for hydrological modelling as simulations of runoff processes are heavily dependent on how realistically the precipitation field is represented both spatially and temporally. Information on large rainfalls also important for many engineering investigations where the design loading conditions of interest are dependent on the characteristics of
daily rainfall. The superior performance of BARRA in representing large rainfall strengthen a case for its use over ERA-Interim where information about extremes is required.

BARRA closely reproduces the frequency of wet days and dry-wet transition probabilities. This evaluation broadly supports BARRA precipitation for its ability to reproduce precipitation statistics at a daily scale. BARRA precipitation could be useful

for assessing variation in the spatial and temporal characteristics of precipitation in a consistent manner that is not influenced by differences in gauge density. In addition, it could potentially be used as a source of data in the central arid zone where AWAP estimates are poor or not available. Considering the limitations in the availability of gauged datasets, BARRA could be considered as a valuable reference dataset for hydro-climatic analysis across the whole of Australia, particularly where

gauging density is low.

The core attraction of the BARRA dataset is the availability of sub-daily precipitation estimates. Such information is not available in the AWAP data set, and the spatial resolution of the estimates is higher than the currently available global reanalysis and satellite datasets. Accordingly, future work will be directed towards an analysis of BARRA estimates of sub-daily precipitation to assess its ability to represent precipitation at finer time scales.

**Code availability**

Codes used for the analysis are available in the supplement.

**Author contributions**

SCA designed the research and performed the analysis. All co-authors provided ideas and feedback following discussions. SCA prepared the paper with contributions from all co-authors..

**Competing interests**

The authors declare that they have no conflict of interest.

**Acknowledgments**

The authors gratefully acknowledge the financial support provided by Seqwater and the Bureau of Meteorology to partially fund the first author's PhD scholarship. We would like to thank colleagues at the Bureau of Meteorology (Peter Steinle, Robert

Smalley and Alex Evans) for discussions at various stages of research. We are also grateful to Dörte Jakob for commenting on early results and for providing feedback on drafts of the manuscript.

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

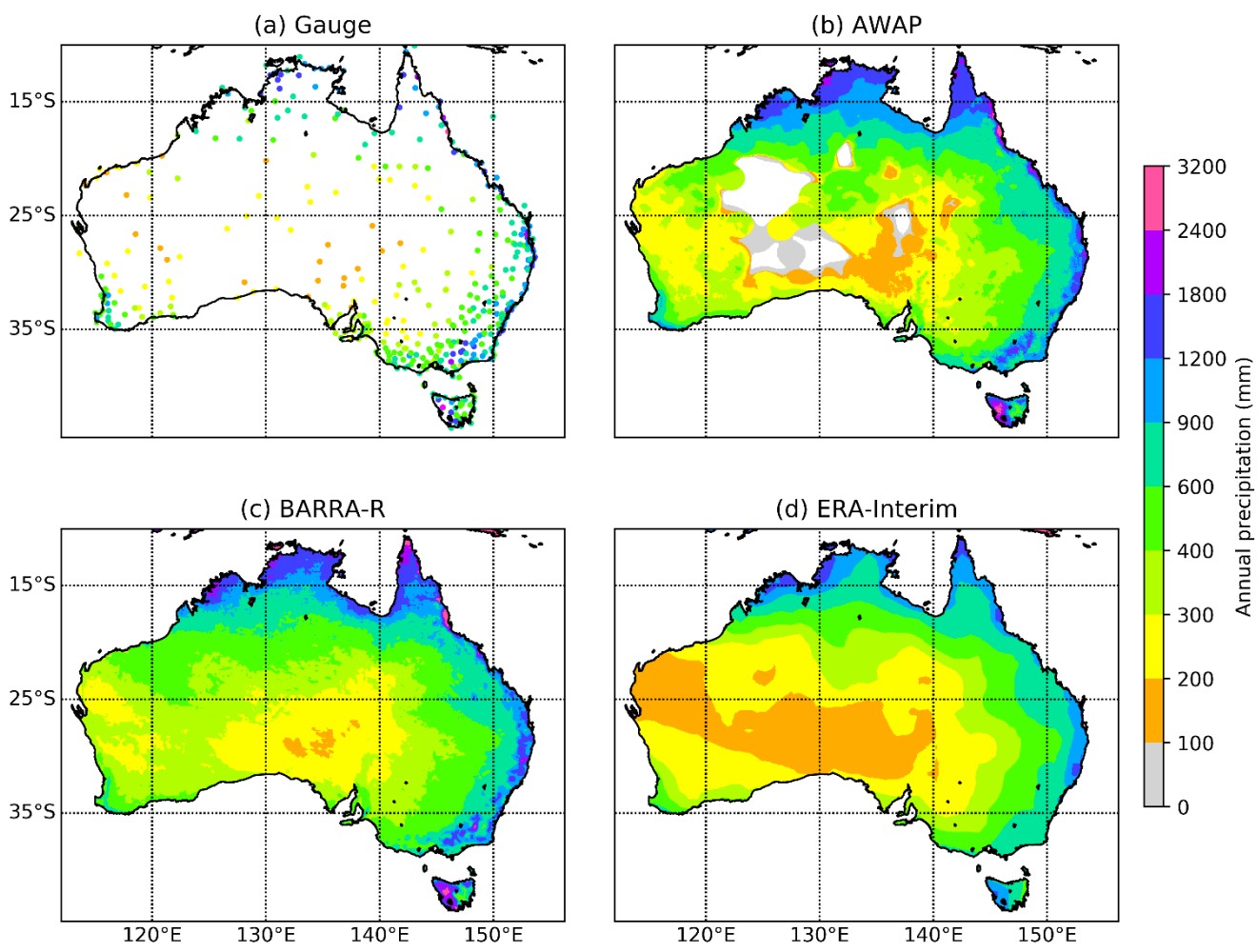

**Figure 1 Average annual precipitation over Australian region (a)Gauge data, (b) AWAP, (c) BARRA, and (d) ERA-Interim. Missing values and ocean are masked in all datasets.**

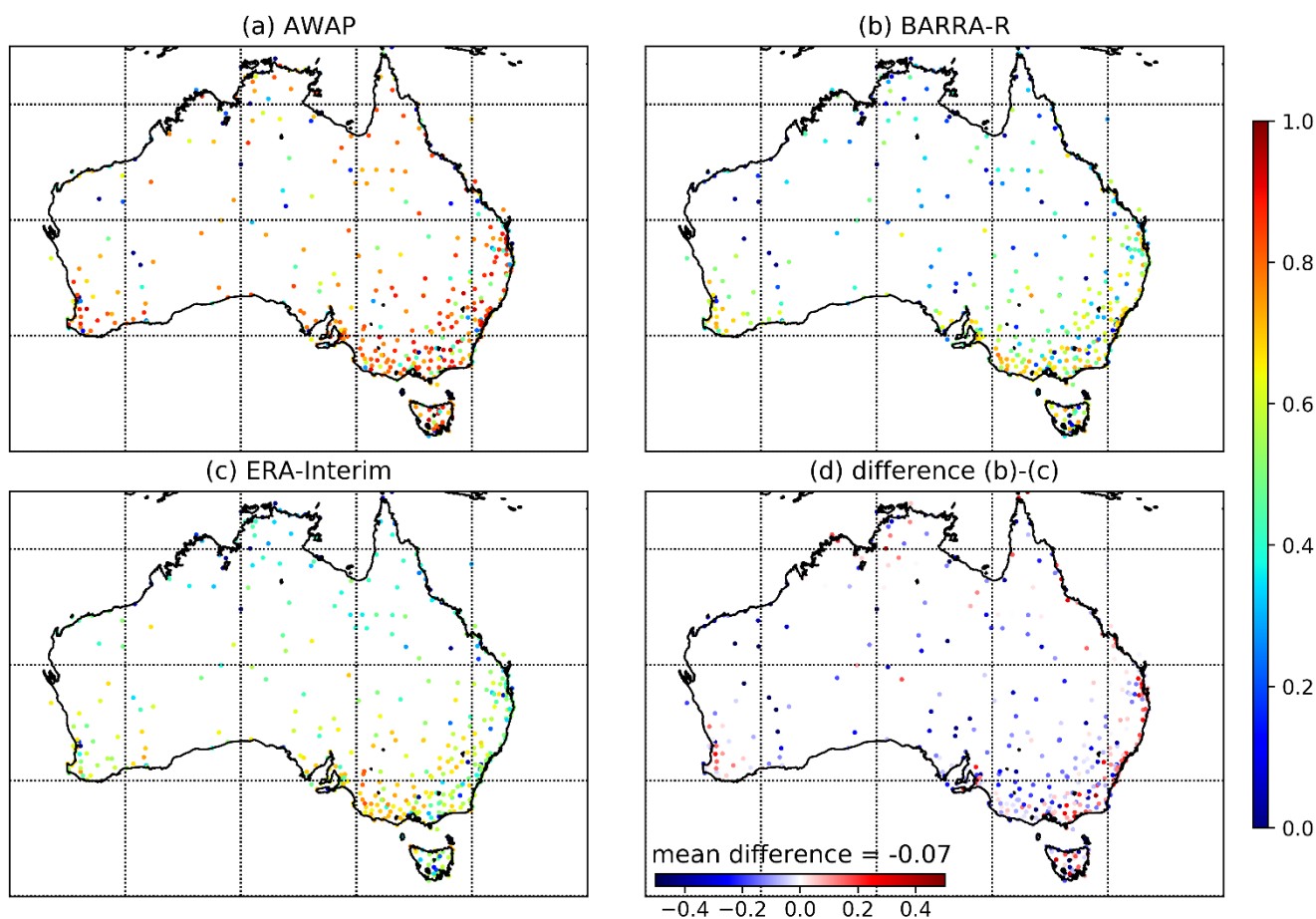

**Figure 2 Modified Kling-Gupta efficiency (KGE') against gauge dataset at daily scale for (a) AWAP, (b) BARRA, (c) ERA-Interim, and (d) difference of KGE' between BARRA and ERA-Interim.**

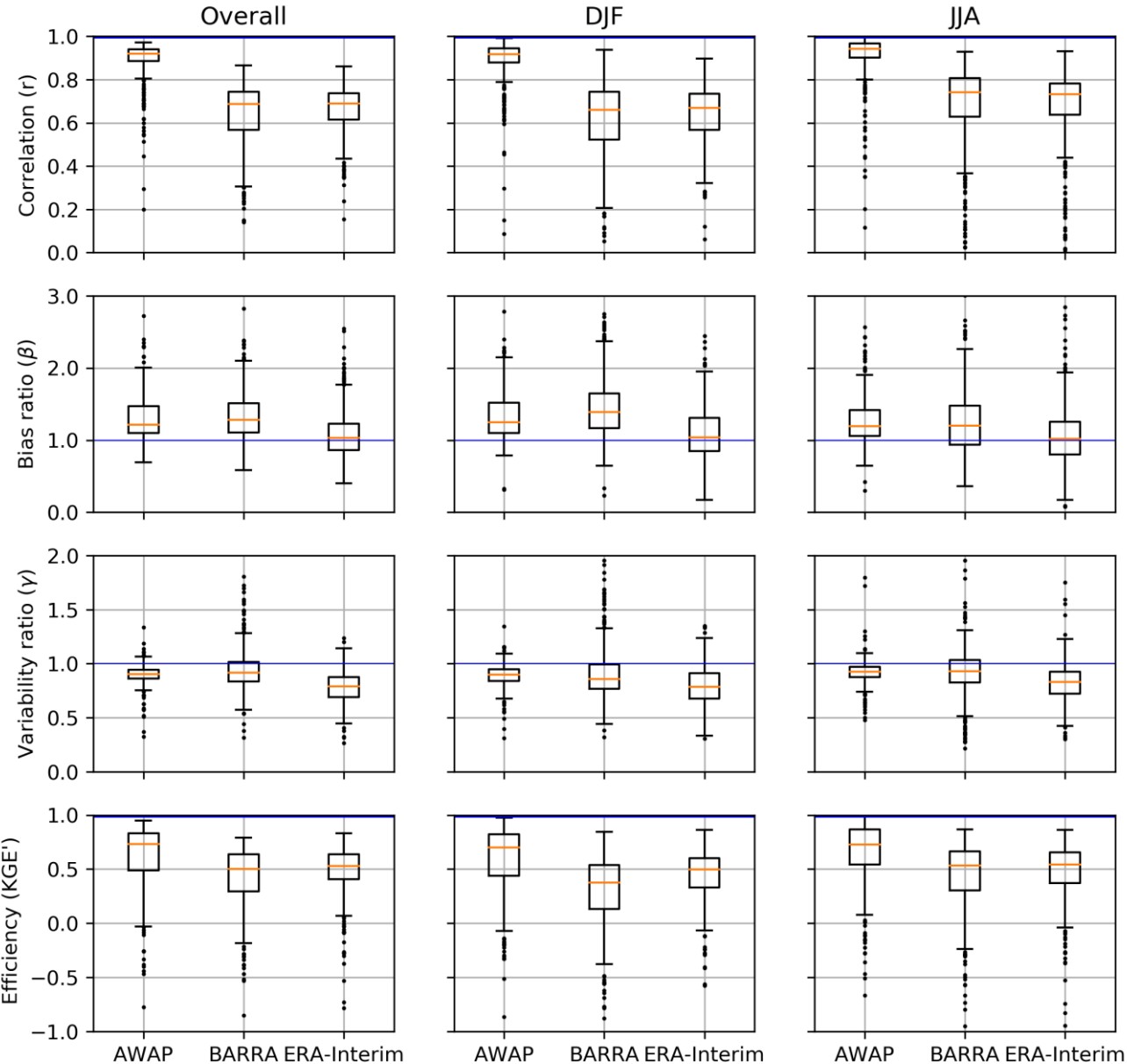

**Figure 3 Boxplot of correlation (r), bias ratio (β), variability ratio (γ), and modified Kling-Gupta Efficiency (KGE') between daily precipitation estimates from gridded datasets and observations at gauge locations. Columns represent overall data, summer (DJF), and winter (JJA) respectively. Each box extends from first to third quartile, medians are marked in each box, and whisker extends to furthest point or 1.5 times the interquartile range whichever is closer. The blue horizontal line represents the best value.**

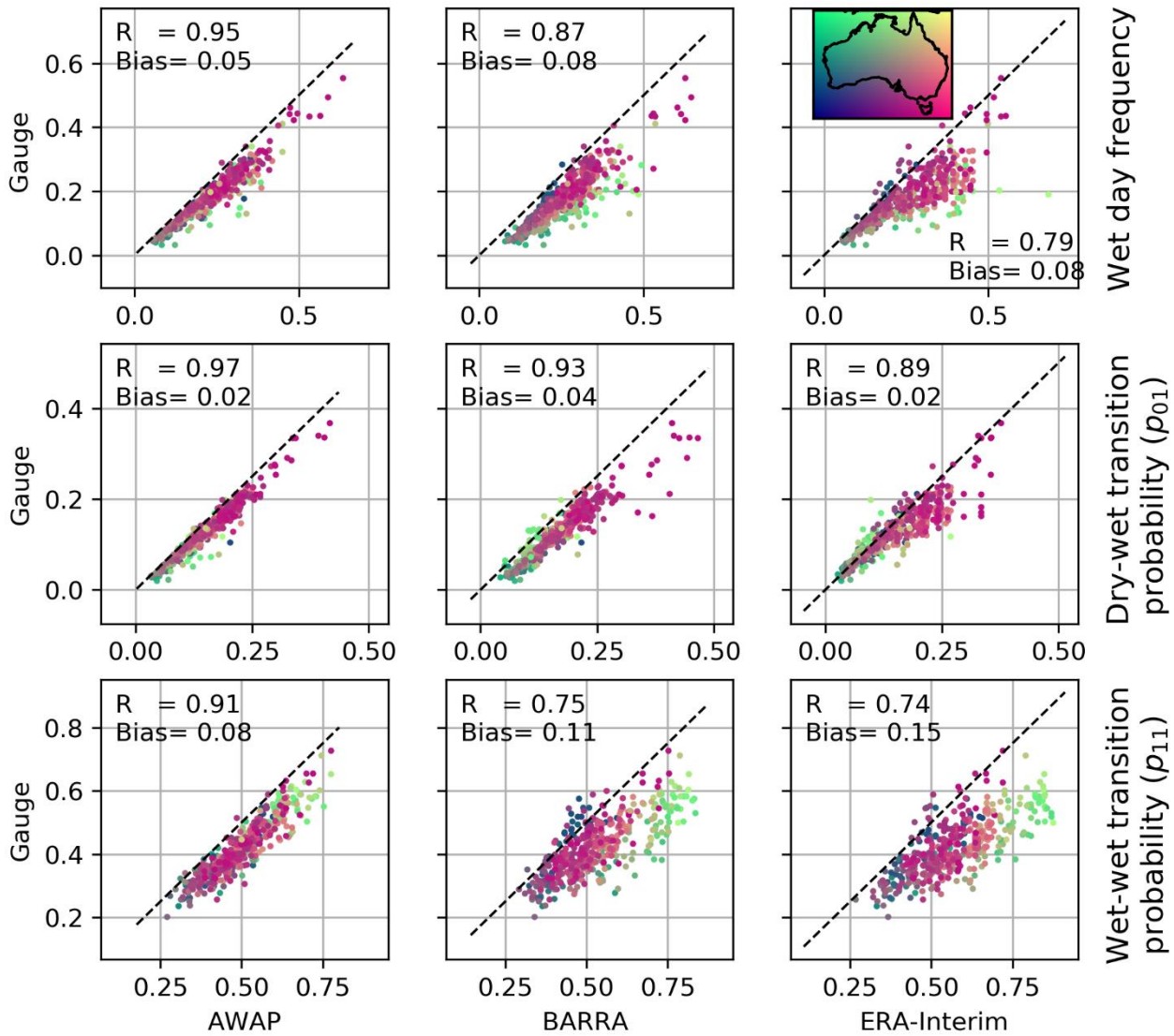

**Figure 4 Wet day frequency estimates from gridded datasets (AWAP, BARRA, and ERA-Interim) and observations at gauge locations (first row). Transition probabilities: dry-wet ($p_{01}$, second row), and wet-wet ($p_{11}$, third row). Colour of scatter indicates the location of the station.**

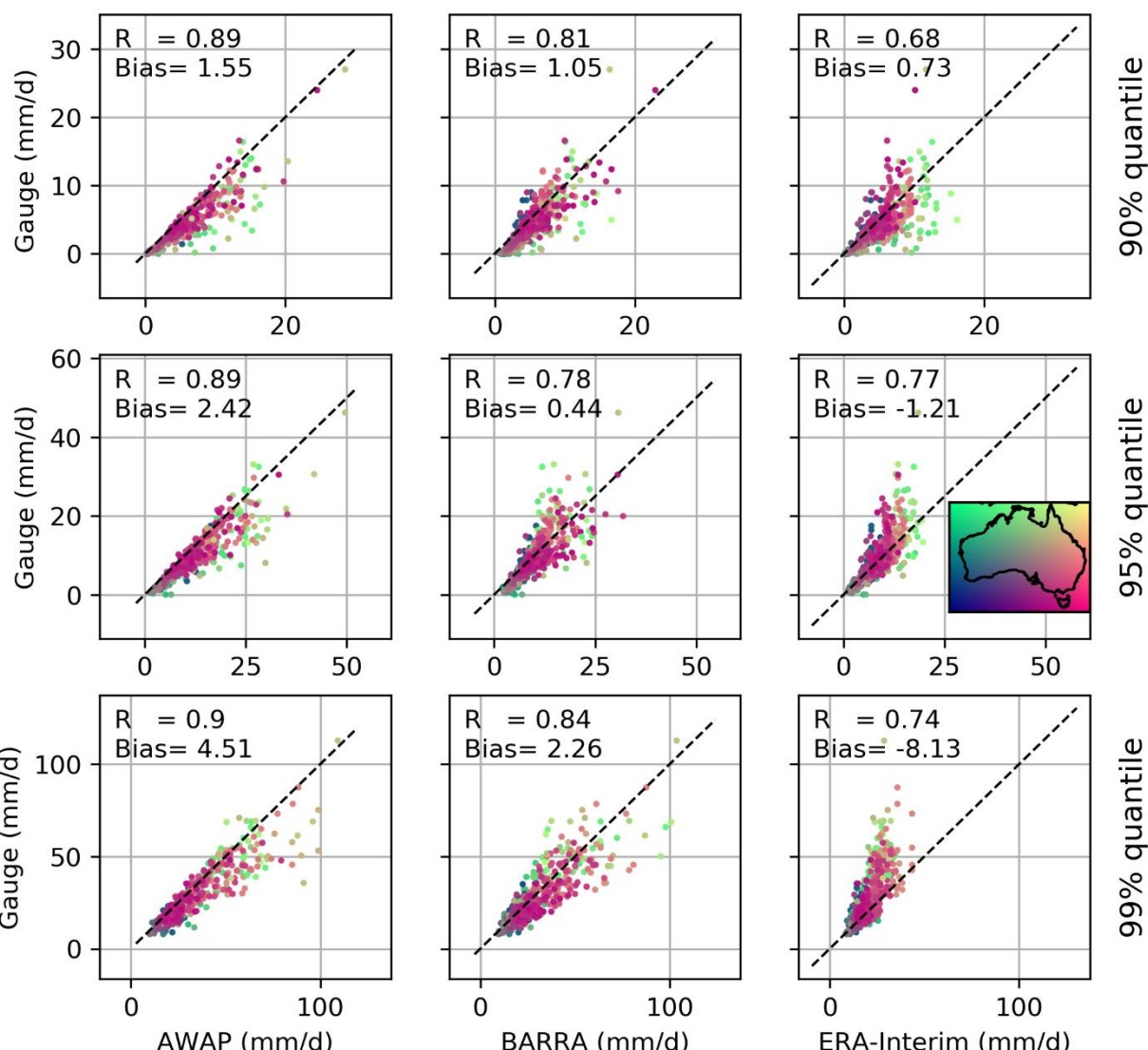

**Figure 5** Quantile estimates (mm/day; 90, 95 and 99% in the top, middle and bottom row respectively) from gridded datasets (AWAP, BARRA, and ERA-Interim) and observations at gauge locations. The colour of the scatter indicates the location of the stations.

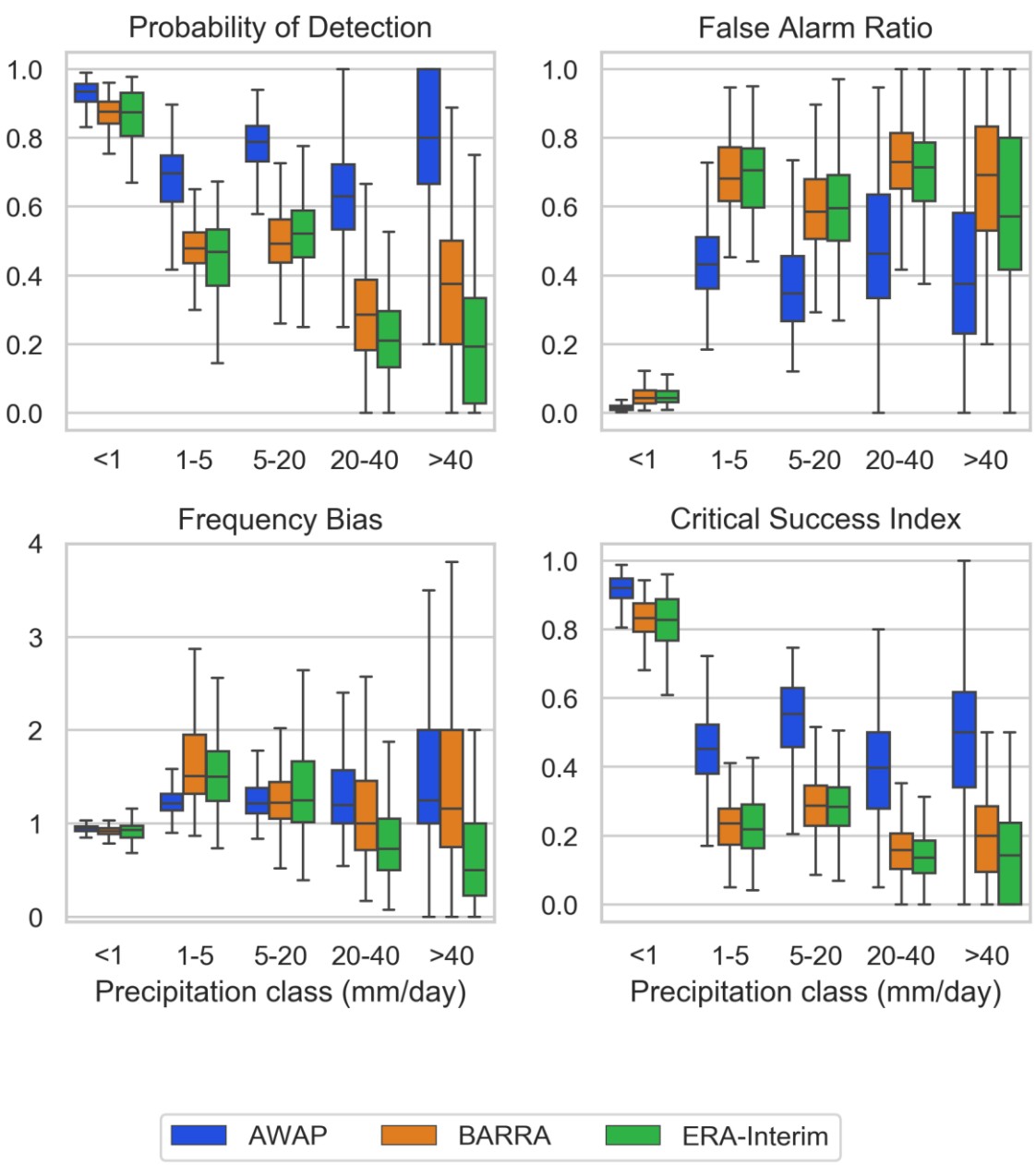

**Figure 6 Boxplot of categorical performance indices (Probability of detection, false alarm ratio, frequency bias , and critical success index) calculated against gauge data for five classes of rainfall intensity. Each box extends from first to third quartile, medians are marked in each box, and whisker extends to furthest point or 1.5 times the interquartile range whichever is closer.**

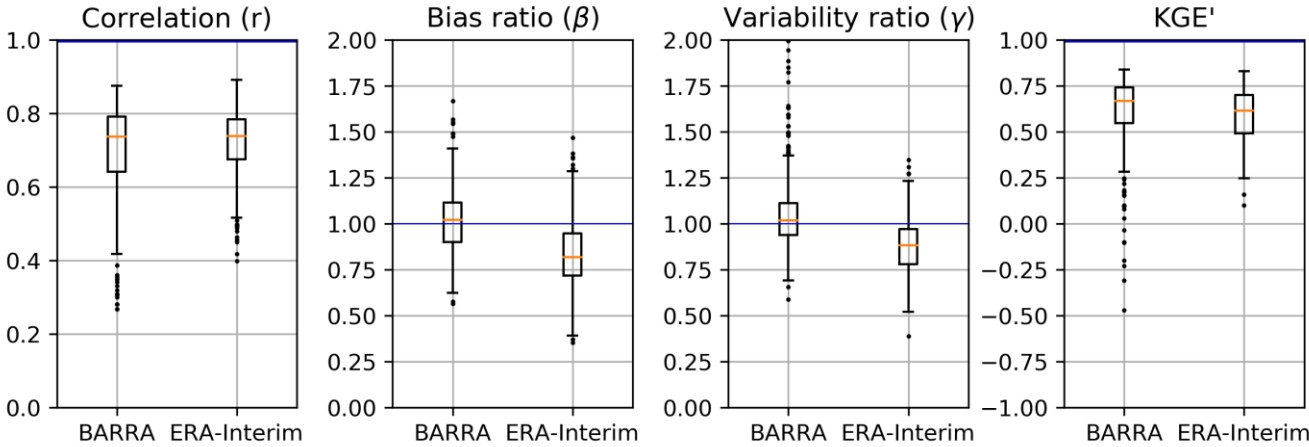

**Figure 7 Boxplot of correlation (r), bias ratio (β), variability ratio (γ), and modified Kling-Gupta Efficiency (KGE') against AWAP dataset. Each box extends from first to third quartile, medians are marked in each box, and whisker extends to furthest point or 1.5 times the interquartile range whichever is closer.**

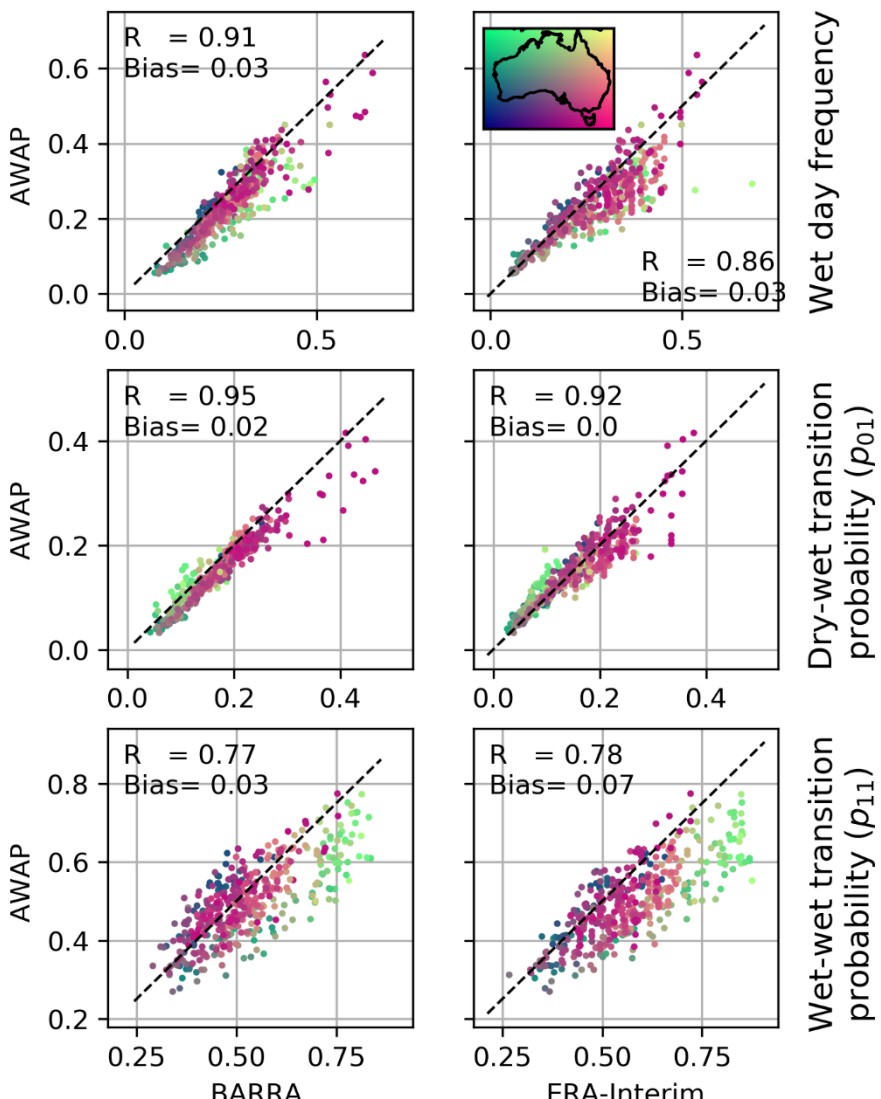

**Figure 8 Wet day frequency estimates from reanalysis datasets (BARRA, and ERA-Interim) and AWAP at gauge locations (first row). Transition probabilities: dry-wet ($p_{01}$, second row), and wet-wet ($p_{11}$, third row). Colour of scatter indicates the location of the AWAP grid.**

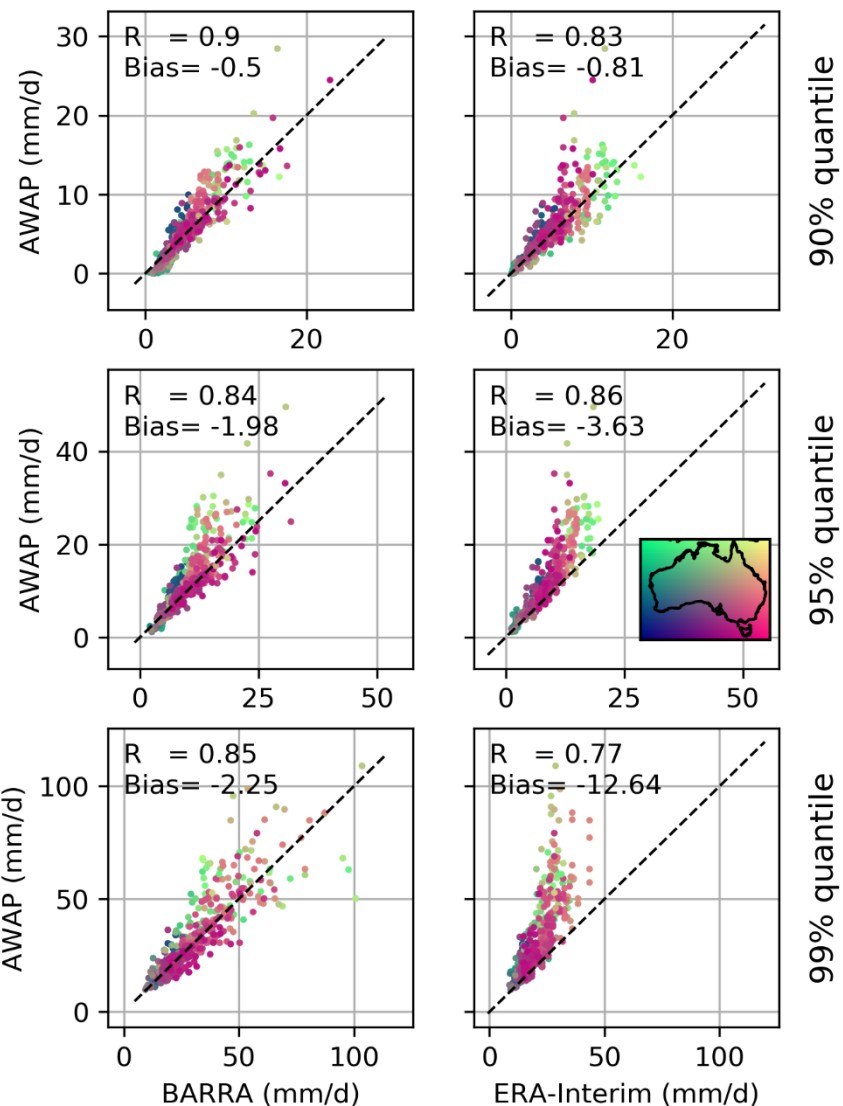

**Figure 9 Quantile estimates (mm/day; 90, 95 and 99%, in the top, middle and bottom row respectively) from reanalysis datasets (BARRA, and ERA-Interim) and AWAP at gauge locations. The colour of the scatter indicates the location of the AWAP grid.**

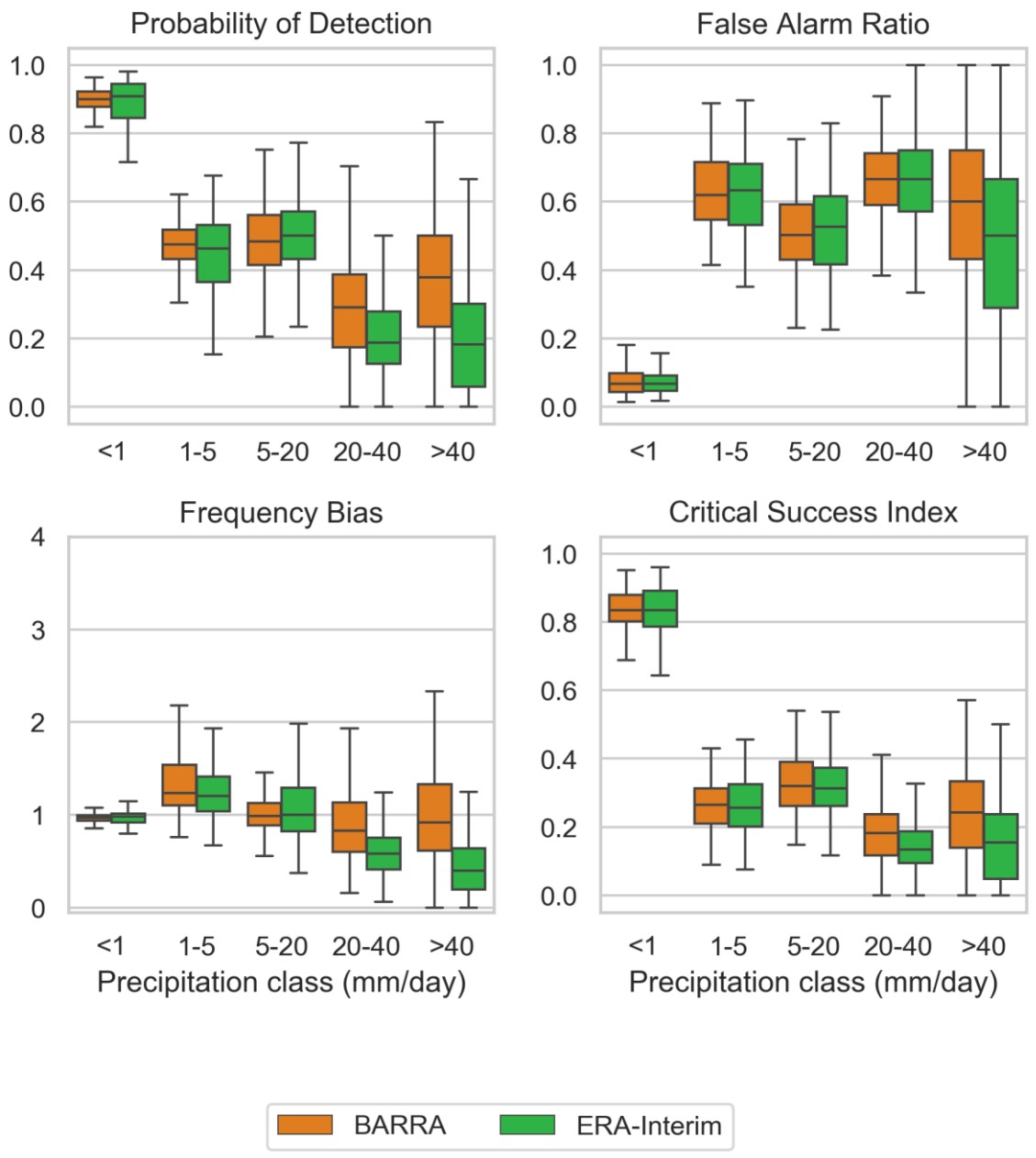

**Figure 10 Boxplot of categorical performance indices (Probability of detection, false alarm ratio, frequency bias , and critical success index) calculated against AWAP data for five classes of rainfall intensity. Each box extends from first to third quartile, medians are marked in each box, and whisker extends to furthest point or 1.5 times the interquartile range whichever is closer.**

**Table 1 Overview of gridded precipitation datasets used in this study**

| Name | Details | Data Source | Spatial coverage and resolution | Temporal coverage and resolution | Reference |
|------|---------|-------------|--------------------------------|----------------------------------|-----------|
| BARRA-R | Bureau of Meteorology Atmospheric high-resolution Regional Reanalysis for Australia (http://www.bom.gov.au/research/projects/reanalysis/) | Regional reanalysis | Australasian region (65 to 196.9° E, -65 to 19.4° N), ~12km | 1990-February 2019, Hourly | (Su et al., 2019) |
| ERA-Interim | European Centre for Medium-range Weather Forecasts ReAnalysis Interim (https://www.ecmwf.int/en/forecasts/datasets/reanalysis-datasets/era-interim) | Global reanalysis | Global, ~80km | 1979-present, 3-hourly | (Dee et al., 2011) |
| AWAP | Australian Water Availability Project (http://www.csiro.au/awap/) | Gauge interpolated | Australia land area, 5km | 1900 – present, Daily | (Jones et al., 2009) |

5    **Table 2 Median values of KGE' and its components grouped for broad Koppen-Geiger climate categories for Australia. The number in parenthesis indicates the number of stations in the climatic zones. Value in bold represent the best score in each group**

| | | Tropical (50) | | | Arid (125) | | | Temperate (266) | | |
|---|---|---|---|---|---|---|---|---|---|---|
| | | AWAP | BARRA | ERA Interim | AWAP | BARRA | ERA Interim | AWAP | BARRA | ERA Interim |
| Against Gauge | Correlation (R) | **0.88** | 0.42 | 0.55 | **0.91** | 0.60 | 0.65 | **0.93** | 0.72 | 0.71 |
| | Bias ratio ($\beta$) | 1.24 | 1.28 | **0.97** | 1.22 | 1.35 | **1.01** | 1.20 | 1.22 | **1.06** |
| | Variability ratio ($\gamma$) | **0.87** | 1.15 | 0.61 | **0.90** | 0.85 | 0.86 | 0.91 | **0.93** | 0.79 |
| | KGE' | **0.67** | 0.27 | 0.35 | **0.72** | 0.41 | 0.53 | **0.74** | 0.58 | 0.56 |

| | Metric | | | | | | |
|---|---|---|---|---|---|---|---|
| **Against AWAP** | Correlation (R) | 0.49 | **0.63** | 0.66 | **0.67** | **0.77** | 0.76 |
| | Bias ratio (β) | **0.99** | 0.76 | **1.08** | 0.83 | **0.99** | 0.82 |
| | Variability ratio ($\gamma$) | 1.35 | **0.72** | **0.95** | 0.94 | **1.02** | 0.88 |
| | KGE' | 0.38 | **0.43** | **0.60** | **0.60** | **0.71** | 0.64 |

**Table 3 Summary of performance of BARRA precipitation**

| Metric | Point-to-grid | Grid-to-grid |
|---|---|---|
| KGE' and its components | • Performance varies spatially, showing better scores in temperate regions than in the tropical and arid region<br>• Variability is well captured | • Exhibits less bias in total rainfall than ERA-Interim<br>• Overestimates variability in the tropics only |
| Quantiles | • High quantiles (90, 95 and 99%) closely represent high rainfalls and are unbiased. | • Similar pattern as for point-to-grid<br>• Better representation of higher quantiles than ERA-Interim |
| Wet day frequency and transition probabilities | • Frequency of wet days and wet-wet transition probabilities are over-estimated<br>• Dry-wet transition probabilities are well reproduced | • Similar pattern as for point-to-grid with improved correlation and reduced bias. |
| Categorical metrics | • Both POD and FAR increase with threshold<br>• Frequency bias is on average higher for light/moderate than low and heavy rainfalls. | • Improvement along all metrics with respect to ERA-Interim<br>• Light rainfalls are over-estimated<br>• Large rainfalls are poorly captured yet are better than ERA-Interim |