# Peer review of "An evaluation of daily precipitation from a regional atmospheric reanalysis over Australia"

_Hydrology and Earth System Sciences, 2018_

## Referee Comment (RC1) · Anonymous Referee #1 · 18 Mar 2019

Dear authors,

The manuscript "An evaluation of daily precipitation from atmospheric reanalyses over Australia" aims at comparing the new reanalysis precipitation dataset BARRA with ERA-Interim over Australia using in-situ rainfall data (point-to-grid analysis) and AWAP dataset (grid-to-grid analysis) as benchmarks. I do believe that the paper reads very well, it is properly structured and addresses a relevant topic of uttermost importance. The authors showed that the new dataset BARRA tends to outperform in most of the case ERA-Interim, while provides lower performances when compared to the AWAP dataset. In my opinion, I found the comparison of BARRA with only 1 reanalysis dataset not enough to justify a possible publication in HESS. My main comments are:

1. The purpose of the current study is to document the performance of the BARRA

dataset at a daily scale and to provide a comparative analysis of its strengths and limitations relative to other available datasets. However, only ERA-Interim is used as a comparison. Why did the authors decide to use only 1 dataset for comparison? Why the choice of using another reanalysis dataset (ERA-Interim) and not other based purely on satellite product (e.g. PERSIANN) or corrected satellite (e.g. PERSIANN-CDR)? I believe the manuscript (and the comparison) will benefit with the inclusion of additional recent and well-known datasets (e.g. CHIRPS, MSWEPv2.1, SM2RAIN ASCAT, CMORPH-CRT), or other reanalysis datasets (e.g. JRA-55, NCEP-CFSR, PFD, or WFEDEI GPCC) for comparison. Obviously, I am not suggesting to include several datasets in this analysis, but the comparison with 3 or 4 more datasets will definitely strengthen the impact of this research and manuscript.

2. The authors first mentioned that "The accuracy at a daily scale provides us with an important benchmark as it is applicable to many hydrological applications and also forms the basis for further examination at finer timescales". However, the author then contradicted themselves concluding that "The core attraction of the BARRA dataset is the availability of sub-daily precipitation estimates. Such information is not available in the AWAP data set, and the spatial resolution of the estimates is higher than the currently available global 20 reanalysis and satellite datasets". In fact, in hydrological application at large scale (which is the case for the BARRA dataset due to a spatial resolution of 36km) daily time scale is most used temporal resolution. For this rea-son, as end-user, I would select the AWAP dataset as input for a large scale model as the resolution is higher and more appropriate to represent complex topographies. Beside the scientific interest in comparing different precipitation datasets, why some-one should use BARRA if AWAP is already providing excellent performances at higher spatial resolution?

3. Results and discussions of grid-to-grid analysis are very brief and conclusions are somehow similar to the point-to-grid analysis in which BARRA gives better results than ERA: What is the additional value of including such analysis? It would be better to

include more dataset for comparison (see the first point) and run only point-to-grid analysis.

4. How the different spatial resolution of the reanalysis dataset and interpolation method of the in-situ gauges can affect the results of this analysis?

5. The methodology is quite straightforward and based on existing approaching for comparing distributed precipitation dataset. Besides the comparison of different datasets over Australia using different performance measures which one is the main research innovation of this paper?

6. From Figure 2.d it is difficult to assess where BARRA is performing (on average) better than ERA-Interim. From my point of view, ERA-Interim shows overall higher KGE values than BARRA (blue points). I suggest the authors to estimate the average (and standard deviation) of the values in figure 2.d to see which dataset provides higher KGE.

7. Lines 18-20 page 7 "Despite having a slightly lower correlation compared to ERA-Interim, the variability of the rainfall is better captured by the BARRA dataset" I do not agree with the authors. From figure 3 it looks that ERA tends to outperform BARRA in almost all the considered performance measures. Also, how the authors can say that rainfall is better captured by BARRA dataset if an aggregated index (KGE) is used?

8. Line 14, page 8 "The spatial pattern, however, is similar for all datasets." Not really. It looks to me that spatial pattern is different from figure 4. Are the authors referring only to the spatial pattern of BARRA and ERA-Interim?

---

## Short Comment (SC1) · 22 Mar 2019

We do appreciate the detailed review by Anonymous Referee 1 and her/his overall positive impression of our work. Here we briefly clarify some comments in order to stimulate further discussion.

The reviewer's main comment (point 1) is that the comparison of the performance against multiple datasets (global reanalysis, satellite product, or corrected satellite products) would strengthen the impact of the research. While we agree that a comparative analysis covering more datasets is instructive, this study focuses on an evaluation of the new regional reanalysis dataset BARRA and does not aim to instruct users on which of the wide-ranging precipitation products should be used over Australia. The

choice of AWAP and ERA-Interim datasets are well-suited to addressing the study objectives as AWAP is a daily, high-resolution gridded dataset which is widely accepted as being the best synthesis of gauged observations, and ERA-Interim is known to perform well when compared to other rainfall products in the Australian region (Peña-Arancibia et al., 2013). Assessing the skill of BARRA to estimate precipitation is of most interest as precipitation observations are not assimilated in BARRA analysis.

Furthermore, the reviewer questions, in point 2, the evaluation of BARRA at a daily timescale given the potential utility of its sub-daily estimates. We do agree that the sub-daily estimates have the potential to provide great value to users, though there are two points that underscore the importance of first looking at its ability to represent daily performance. First, the accuracy of AWAP suffers greatly from the uneven distribution of gauge networks (see Fig 1b), and the potential advantage of BARRA lies in its ability to complement the existing dataset at the regions where gauged observations are sparse (especially in the semi-arid regions). Second, any reliance on sub-daily estimates is necessarily dependent on its ability to correctly represent daily rainfalls, so this is seen as a necessary first step towards examining sub-daily behaviour (information which is not available from the AWAP product). This will inform our future work of conducting sub-daily evaluation, which requires more sophisticated methods and involves more complicated results.

We would be pleased to address the Reviewer's concerns more comprehensively once the discussion period has closed.

Reference

Peña-Arancibia, J. L., van Dijk, A. I. J. M., Renzullo, L. J. and Mulligan, M.: Evaluation of Precipitation Estimation Accuracy in Reanalyses, Satellite Products, and an Ensemble Method for Regions in Australia and South and East Asia, J. Hydrometeorol., 14(4), 1323–1333, doi:10.1175/JHM-D-12-0132.1, 2013.

607, 2019.

---

## Referee Comment (RC2) · Korbinian Breinl (Referee) · 26 Mar 2019

Dear authors,

The present article deals with the evaluation of a new reanalysis dataset called BARRA with other gridded datasets (ERA-Interim and AWAP) and rain gauge observations, for Australia. This is a well written paper, which easy to read. At this stage I think however that some improvements are needed for the publication in HESS.

1. I wonder if the results of the mean precipitation (Figure 1 and related text in the results section) could be presented in a different way – besides the four maps that are useful for sure – to better capture the changes in spatial variability. Maybe the authors have a good idea for a plot. Also, I wonder if cutting off the over-sea precipitation

for BARRA and ERA-Interim would help to better read the map, meaning taking the over-land precipitation as the lowest common denominator.

2. I think it would be good to look into spatial correlations. As far as I can see you have not looked into them, although they are relevant. Is there a particular reason for not taking them into account?

3. I do not fully understand the purpose of the BARRA dataset, at least not in the context of the article, which is focusing on daily values. Considering that the AWAP dataset is superior to BARRA (superior at daily timescale), why would I use BARRA (when sub-daily is not the topic)?

4. What exactly is the added value of comparing the gridded data among each other, without the point rainfall? I would like to try to understand the motivation behind it, isn't the comparison with the point rainfall sufficient enough?

5. In that context (point 4), it would be good to see what has already been done in terms of evaluations of gridded rainfall vs. point rainfall (and also grid rainfall to other grid rainfall), as conducted in this study, means I would like to see a more comprehensive literature review. A brief review of such evaluations helps the reader to better understand the implications of the present study. For example, what are the pros and cons of ERA-Interim according to other studies, and what has been concluded in this article? Again, I miss a bit the ability to generalize from the results from this article.

6. I would appreciate a final overview that summarizes the results, ideally in the format of a table. The table could contain for example (i) general information for each dataset (time period, spatial resolution, URL to obtain data), (ii) metrics for evaluation, (iii) performance of each metric etc. The table would make it much easier to get a good overview of the results. And, even better, another column could add some information on each dataset or similar datasets (similar in terms of how they were processed) and some results from other studies if applicable (see comment above).

7. Reviewer #1 asked for innovation. This is not my main concern as long as I can draw general conclusions from this paper as a reader for my own studies (also outside Australia), but at this stage I miss it a bit. The article tries to explain the results to a certain degree, but I think the discussion should be more detailed, and pros and cons of each dataset should be better explained considering how the datasets were generated, considering seasons, resolution etc. I find statements such as "AWAP estimates of point rainfall are higher than the gauged observations" or "ERA-Interim generally performs better in the central arid region, whereas BARRA exhibits better scores in the temperate region", but I miss good explanations why. The table overview I addressed may help to make this generalization easier. Also, I must admit, looking into more re-analysis data (ideally popular datasets) as suggested by Reviewer #1 may make the paper stronger – at least one more prominent dataset maybe?

8. Figure 6. I would add x-axis labels to the upper plots

9. Optional: The code (R, MATLAB etc.) to analyse the rainfall data could be published with the paper. I always encourage to do so, but this is up to the authors of course.

Sincerely, Korbinian Breinl, TU Vienna, Austria

---

## Author Comment (AC1) · 19 Apr 2019

**Response to reviews on the manuscript hess-2018-607 "An evaluation of daily precipitation from atmospheric reanalyses over Australia" by Suwash Chandra Acharya et al.**

We would like to thank anonymous referee (R1) and Korbinian Breinl (R2) for their constructive comments and suggestions on our paper. These comments have greatly helped us identify where we need to improve our description of the overall context and framing of the research. There are also several specific points which we could address to improve the way we have provided and discussed the results. We comment first on a general point regarding the purpose of the work raised by both reviewers, and this is followed by responses to more specific issues raised by the individual reviewers.

**Comments made by both reviewers**

Both reviewers query why we examine the daily performance of BARRA when it is an hourly product, and both reviewers query whether it would be useful to include comparisons with other reanalysis data sets.

We choose to focus our evaluation on daily rainfalls as this time step affords the best means of comparison with gridded data (AWAP) that is well grounded in gauged observations. AWAP is a high-resolution gridded dataset representative of areal rainfalls which is widely accepted as being the best synthesis of gauged daily observations at the 5-25 km resolution (there is no equivalent available dataset for sub-daily rainfalls). We see this as a necessary first step towards examining sub-daily behaviour because any reliance on sub-daily estimates necessarily depends on its ability to correctly represent daily rainfalls. This will inform our future work of conducting sub-daily evaluation, which requires more sophisticated methods and yields more complicated results.

We do include comparison with daily gauged (point) rainfalls and with gridded ERA-interim. We consider daily gauged point rainfalls as these represent the base data on which the AWAP estimates are derived, though as discussed in the paper we would expect there to be differences between point and gridded estimates as the latter data set accounts for some spatial averaging. Our rationale for including ERA-Interim is that it is has been found to be the best performing data set compared to other reanalysis rainfall products in the Australian region (Peña-Arancibia et al., 2013), and it shares similarity with BARRA in that rainfall observations are not assimilated. Importantly, ERA-Interim also provides the boundary condition for BARRA-R simulations and this allows the incremental value of BARRA-R to be assessed. Our focus with these comparisons is solely on the efficacy of the BARRA-R rainfall product, we do not set out to instruct users on the selection of wide-ranging precipitation products that are available across Australia.

Lastly, one additional reason for considering the performance of BARRA-R at a daily time step is that it has the potential to provide more accurate – or at least a credible alternative – estimate of daily rainfalls in regions which are sparsely gauged. There are large areas of Australia with very sparse gauging (see left-hand panel in the figure below), and it may be useful to supplement estimates of rainfalls in these regions with reanalysis products that do not rely on rainfall observations for assimilation. Without observations to inform surface fitting, it is evident that AWAP can yield erroneous results (see right hand panel in the figure below). One potential advantage of BARRA thus lies in its ability to complement the existing datasets in regions where gauged observations are sparse (especially in the semi-arid regions). In this regard, we note that only AWAP grid points that contain rainfall gauges are selected for comparisons in this study.

In the following sections we provide a detailed response to all the remarks raised by the referees. We had posted a short comment earlier to clarify the general concern raised by R1, and we now proceed to address all the comments by listing the reviewers' comments (RC, in blue), our corresponding reply (AR, in black), and proposed modifications in italics.

[Figure]

**Figure showing location of all daily rainfall stations in Australia (left panel) and AWAP estimates that erroneously indicate zero and less than 10mm total rainfalls over a twenty year period (right panel). The left panel is sourced from the Australian Bureau of Meteorology** (http://www.bom.gov.au/climate/data).

**Response to Referee #1**

**Response to general comment**

[RC] The manuscript "An evaluation of daily precipitation from atmospheric reanalyses over Australia" aims at comparing the new reanalysis precipitation dataset BARRA with ERA-Interim over Australia using in-situ rainfall data (point-to-grid analysis) and AWAP dataset (grid-to-grid analysis) as benchmarks. I do believe that the paper reads very well, it is properly structured and addresses a relevant topic of uttermost importance. The authors showed that the new dataset BARRA tends to outperform in most of the case ERA-Interim, while provides lower performances when compared to the AWAP dataset. In my opinion, I found the comparison of BARRA with only 1 reanalysis dataset not enough to justify a possible publication in HESS.

[AR] We thank Referee #1 for acknowledging the relevance and importance of the topic discussed.

**Responses to specific comments**

1.1 The purpose of the current study is to document the performance of the BARRA dataset at a daily scale and to provide a comparative analysis of its strengths and limitations relative to other available datasets. However, only ERA-Interim is used as a comparison. Why did the authors decide to use only 1 dataset for comparison? Why the choice of using another reanalysis dataset (ERA-Interim) and not other based purely on satellite product (e.g. PERSIANN) or corrected satellite (e.g. PERSIANNCDR)? I believe the manuscript (and the comparison) will benefit with the inclusion of additional recent and well-known datasets (e.g. CHIRPS, MSWEPv2.1, SM2RAIN ASCAT, CMORPH-CRT), or other reanalysis datasets (e.g. JRA-55, NCEP-CFSR, PFD, or WFEDEI GPCC) for comparison. Obviously, I am not suggesting to include several datasets in this analysis, but the comparison with 3 or 4 more datasets will definitely strengthen the impact of this research and manuscript.

[AR] As mentioned above, our aim is to evaluate the performance of BARRA dataset rather than provide instructive comments on use of wide-ranging precipitation products. We will provide additional clarification on these points in a revised version of the Introduction. Further, we propose to change the title of the paper to "*An evaluation of daily precipitation from a regional atmospheric reanalysis over Australia*" to better reflect the focus of our paper.

1.2 The authors first mentioned that "The accuracy at a daily scale provides us with an important benchmark as it is applicable to many hydrological applications and also forms the basis for further examination at finer timescales". However, the author then contradicted themselves concluding that "The core attraction of the BARRA dataset is the availability of sub-daily precipitation estimates. Such information is not available in the AWAP data set, and the spatial resolution of the estimates is higher than the currently available global 20 reanalysis and satellite datasets". In fact, in hydrological application at large scale (which is the case for the BARRA dataset due to a spatial resolution of 36km) daily time scale is most used temporal resolution. For this reason, as end-user, I would select the AWAP dataset as input for a large scale model as the resolution is higher and more appropriate to represent complex topographies. Beside the scientific interest in comparing different precipitation datasets, why someone should use BARRA if AWAP is already providing excellent performances at higher spatial resolution?

[AR] Addressed above in general comments

1.3 Results and discussions of grid-to-grid analysis are very brief and conclusions are somehow similar to the point-to-grid analysis in which BARRA gives better results than ERA: What is the additional value of including such analysis? It would be better to include more dataset for comparison (see the first point) and run only point-to-grid analysis.

[AR] We do agree that the results of grid-to-grid analysis are brief. However, the performance based on frequency metrics (wet day frequency, dry-wet transition probability, and wet-wet transition probability) are similar for both grid-to-grid and point-to-grid analysis and leads to similar conclusions. It is thus appropriate to present the similar results in a brief manner. However, KGE' and its components at grid-to-grid evaluation shows that BARRA exhibits superior performance at grid scale (5 km) which is not obvious from point-to-grid analysis. Given the spatial averaging that is implicit in grid-based products it is not expected that similar results would be obtained with point-to-grid comparisons, or with grid-to-grid comparisons at different resolutions (AWAP is 5 km, BARRA-R is 12km, and ERA-Interim is 80km).

1.4 How the different spatial resolution of the reanalysis dataset and interpolation method of the in-situ gauges can affect the results of this analysis?

[AR] The effect of spatial resolution is clearly observed in the comparison of quantiles: coarser datasets are not well suited to representing large rainfall due to averaging over larger area. In addition, the choice of interpolation could also affect the analysis of large rainfalls. In this study, nearest neighbour interpolation is used which preserves the magnitude of precipitation (averaged) over the grid, however, bilinear interpolation is likely to smoothen the precipitation field resulting in higher bias for larger rainfall (Accadia et al., 2003). We will make amendments to the manuscript to better clarify the reasoning for our choice of methods and their comparisons.

1.5 The methodology is quite straightforward and based on existing approaching for comparing distributed precipitation dataset. Besides the comparison of different datasets over Australia using different performance measures which one is the main research innovation of this paper?

[AR] We agree that the methodology in this study is straightforward and based on existing approaches for evaluation of precipitation dataset. However, the precipitation dataset used in this study is new and is the only regional reanalysis product developed for the Australasian region. The results of this study are useful for the potential users of the dataset for hydrometeorological studies in the Australasian region.

1.6 From Figure 2.d it is difficult to assess where BARRA is performing (on average) better than ERA-Interim. From my point of view, ERA-Interim shows overall higher KGE values than BARRA (blue points). I suggest the authors to estimate the average (and standard deviation) of the values in figure 2.d to see which dataset provides higher KGE.

[AR] The summary of difference in KGE will be added in the revised plot (Figure 2d)

1.7 Lines 18-20 page 7 "Despite having a slightly lower correlation compared to ERAInterim, the variability of the rainfall is better captured by the BARRA dataset" I do not agree with the authors. From figure 3 it looks that ERA tends to outperform BARRA in almost all the considered performance measures. Also, how the authors can say that rainfall is better captured by BARRA dataset if an aggregated index (KGE) is used?

[AR] We checked the plot and the corresponding statement. The best value for variability ratio is 1 and the variability ratio in BARRA is closer to 1 compared to ERA-Interim in overall as well as seasonal analyses. We will add the lines representing best values in KGE' as well as its components to add interpretation of the plot (Figure 3).

1.8 Line 14, page 8 "The spatial pattern, however, is similar for all datasets." Not really. It looks to me that spatial pattern is different from figure 4. Are the authors referring only to the spatial pattern of BARRA and ERA-Interim?

[AR] Based on figure 4, there is slight difference in magnitude of the frequencies across datasets. We agree that BARRA and ERA-Interim are very close to each other. But, in terms of spatial pattern, all three datasets behave similarly as observed in spatial plot shown below.

[Figure]

**Figure showing frequency of wet days for AWAP, BARRA, and ERA-Interim at gauge locations used in this study.**

**Response to Referee #2 Korbinian Breinl**

**Response to general comment**

[RC] The present article deals with the evaluation of a new reanalysis dataset called BARRA with other gridded datasets (ERA-Interim and AWAP) and rain gauge observations, for Australia. This is a well written paper, which easy to read. At this stage I think however that some improvements are needed for the publication in HESS.

[AR] Thank you for your comments.

**Responses to specific comments**

2.1 I wonder if the results of the mean precipitation (Figure 1 and related text in the results section) could be presented in a different way – besides the four maps that are useful for sure – to better capture the changes in spatial variability. Maybe the authors have a good idea for a plot.

[AR] The four maps in Figure 1 visually demonstrate the spatial variability in precipitation across datasets. However, in order to better evaluate the changes in spatial variability, we propose to compute a correlation measure between BARRA-R annual rainfalls and AWAP, and similarly between ERA-Interim & gauged point rainfalls and AWAP. We will include these comparisons in the body of the text, or if useful we will add a row of scatter plots to Figure 1 that illustrate the nature of the spatial correlations involved.

Also, I wonder if cutting off the over-sea precipitation for BARRA and ERA-Interim would help to better read the map, meaning taking the over-land precipitation as the lowest common denominator.

[AR] Our original logic for retaining this was two-fold: to visualize spatial richness in the datasets, and to show that BARRA extends across ocean surrounding Australia which is not the case for gauged measurements and AWAP. However, we are happy to mask the information shown on these plots so they cover the same physical domain.

2.2 I think it would be good to look into spatial correlations. As far as I can see you have not looked into them, although they are relevant. Is there a particular reason for not taking them into account?

[AR] The correlation coefficient was computed for each station and their spatial distribution was visualised. It was not shown in the manuscript as the spatial distribution of correlation coefficient was very close to spatial distribution of KGE'. We have not looked further into spatial correlation, though as responded in 2.1, the new correlation analysis will explicitly demonstrate the relative ability of the datasets to preserve spatial correlation.

2.3 I do not fully understand the purpose of the BARRA dataset, at least not in the context of the article, which is focusing on daily values. Considering that the AWAP dataset is superior to BARRA (superior at daily timescale), why would I use BARRA (when sub-daily is not the topic)?

[AR] We address the choice of daily scale and its utility in the general response above.

2.4 What exactly is the added value of comparing the gridded data among each other, without the point rainfall? I would like to try to understand the motivation behind it, isn't the comparison with the point rainfall sufficient enough?

[AR] As noted earlier, BARRA represent area-average rainfall: while gauged data represents the best estimate of rainfall at a point location, AWAP serves as the best available estimate of areal rainfall. We think it is useful to undertake both grid-to-point and grid-to-grid comparisons as both types of estimates are likely to contain biases due to differences in spatial averaging, and the levels of measurement support also differs between the different products. The two analyses thus offer different insights to the quality of the model estimates. It is worth noting that in terms of performance, KGE' and its components at grid-to-grid evaluation shows that BARRA has superior performance at grid scale (5 km) which is not obvious from point-to-grid analysis.

2.5 In that context (point 4), it would be good to see what has already been done in terms of evaluations of gridded rainfall vs. point rainfall (and also grid rainfall to other grid rainfall), as conducted in this study, means I would like to see a more comprehensive literature review. A brief review of such evaluations helps the reader to better understand the implications of the present study. For example, what are the pros and cons of ERA-Interim according to other studies, and what has been concluded in this article? Again, I miss a bit the ability to generalize from the results from this article.

[AR] We will add further details of relevant literature that contain grid-to-grid and point-to-grid evaluations. We will include reference to further studies that document the performance of ERA-Interim (especially in Australian region) where such studies are comparable, and we will also include reference to studies that evaluate the Unified Model on which the BARRA-R estimates are derived.

2.6 I would appreciate a final overview that summarizes the results, ideally in the format of a table. The table could contain for example (i) general information for each dataset (time period, spatial resolution, URL to obtain data), (ii) metrics for evaluation, (iii) performance of each metric etc. The table would make it much easier to get a good overview of the results. And, even better, another column could add some information on each dataset or similar datasets (similar in terms of how they were processed) and some results from other studies if applicable (see comment above).

[AR] We will add a table with information on datasets and summary of performance in the revised manuscript (something based on the information presented below).

Table of datasets

| Name | Details | Data Source | Spatial coverage and resolution | Temporal coverage and resolution | Reference |
|---|---|---|---|---|---|
| BARRA | Bureau of Meteorology Atmospheric high-resolution Regional Reanalysis for Australia | Regional reanalysis | Australasian region (65 to 196.9° E, -65 to 19.4° N), ~12km | 1990[1]-present, Hourly | (Su et al., 2018) |
| ERA-Interim | European Centre for Medium-range Weather Forecasts ReAnalysis Interim, https://www.ecmwf.int/en/forecasts/datasets/reanalysis-datasets/era-interim | Global reanalysis | Global, ~80km | 1979-present, 3-hourly | (Dee et al., 2011) |
| AWAP | Australian Water Availability Project | Gauge interpolated | Australia land area, 5km | 1900 – present, Daily | (Jones et al., 2009) |

[1] Planned to go back to 1990. Presently available from 2004-present

| Evaluation metric | Performance | Remarks |
|---|---|---|
| KGE' | • Correlation: similar correlation across reanalysis; lesser difference at grid scale

• Bias: positive bias at point scale for reanalysis and AWAP. BARRA, at grid scale, provides unbiased estimate.

• Variability is well captured by BARRA at both point and grid scale | (We will add remarks regarding the results which will support detailed discussion) |
| Wet day frequency and transition probabilities | More wet days on all datasets | Inherent property of point vs areal estimate |
| Quantiles | Better representation of higher quantiles | High resolution renalaysis better represents extremes (Isotta et al 2014) |
| Categorical metrics | High hit rate and high false alarm | |

2.7 Reviewer #1 asked for innovation. This is not my main concern as long as I can draw general conclusions from this paper as a reader for my own studies (also outside Australia), but at this stage I miss it a bit. The article tries to explain the results to a certain degree, but I think the discussion should be more detailed, and pros and cons of each dataset should be better explained considering how the datasets were generated, considering seasons, resolution etc. I find statements such as "AWAP estimates of point rainfall are higher than the gauged observations" or "ERA-Interim generally performs better in the central arid region, whereas BARRA exhibits better scores in the temperate region", but I miss good explanations why. The table overview I addressed may help to make this generalization easier. Also, I must admit, looking into more reanalysis data (ideally popular datasets) as suggested by Reviewer #1 may make the paper stronger – at least one more prominent dataset maybe?

[AR] Thank you for acknowledging the relevance of the study. We agree that significance of this paper is enhanced by its ability to provide general conclusions useful for other studies, different data sets, or study areas. In that regard, we will revisit the discussion section and explain the physical basis of the results further in order to aid generalization of the research beyond current study area (for example, modelling convective rainfalls remains very challenging at these spatial scales, and the prevalence for these rainfall mechanisms vary with location). In addition, the table generated in responding comment 2.6 will further help explain the results.

As mentioned in the general response, our focus is on evaluating the new BARRA-R reanalysis dataset rather than providing instruction to users on selection of available precipitation datasets. Adoption of a changed title (as discussed in response to 1.1 above) should better reflect the intent of the paper.

2.8 Figure 6. I would add x-axis labels to the upper plots

[AR] Agreed, we will add the x-axis labels in the revised plot.

2.9 Optional: The code (R, MATLAB etc.) to analyse the rainfall data could be published with the paper. I always encourage to do so, but this is up to the authors of course.

[AR] We will identify the codes that we consider most relevant for general use. We will upload the codes as a public GitHub repository and add the corresponding link in the revised manuscript under the sub-section "Data Availability".

**References**

Accadia, C., Mariani, S., Casaioli, M., Lavagnini, A. and Speranza, A.: Sensitivity of Precipitation Forecast Skill Scores to Bilinear Interpolation and a Simple Nearest-Neighbor Average Method on High-Resolution Verification Grids, Weather Forecast., 18(5), 918–932, doi:10.1175/1520-0434(2003)018<0918:SOPFSS>2.0.CO;2, 2003

Isotta, F. A., Vogel, R. and Frei, C.: Evaluation of European regional reanalyses and downscalings for precipitation in the Alpine region, Meteorol. Zeitschrift, 24(1), 15–37, doi:10.1127/metz/2014/0584, 2014.

Peña-Arancibia, J. L., van Dijk, A. I. J. M., Renzullo, L. J. and Mulligan, M.: Evaluation of Precipitation Estimation Accuracy in Reanalyses, Satellite Products, and an Ensemble Method for Regions in Australia and South and East Asia, J. Hydrometeorol., 14(4), 1323–1333, doi:10.1175/JHM-D-12-0132.1, 2013

---

## Author Response (AR1)

**Response to reviews on the manuscript hess-2018-607 "An evaluation of daily precipitation from atmospheric reanalyses over Australia" by Suwash Chandra Acharya et al.**

We would like to thank anonymous referee (R1) and Korbinian Breinl (R2) for their constructive comments and suggestions on our paper. These comments have greatly helped us identify where we need to improve our description of the overall context and framing of the research. There are also several specific points which we could address to improve the way we have provided and discussed the results. We comment first on a general point regarding the purpose of the work raised by both reviewers, and this is followed by responses to more specific issues raised by the individual reviewers.

**Comments made by both reviewers**

Both reviewers query why we examine the daily performance of BARRA when it is an hourly product, and both reviewers query whether it would be useful to include comparisons with other reanalysis data sets.

We choose to focus our evaluation on daily rainfalls as this time step affords the best means of comparison with gridded data (AWAP) that is well grounded in gauged observations. AWAP is a high-resolution gridded dataset representative of areal rainfalls which is widely accepted as being the best synthesis of gauged daily observations at the 5-25 km resolution (there is no equivalent available dataset for sub-daily rainfalls). We see this as a necessary first step towards examining sub-daily behaviour because any reliance on sub-daily estimates necessarily depends on its ability to correctly represent daily rainfalls. This will inform our future work of conducting sub-daily evaluation, which requires more sophisticated methods and yields more complicated results.

We do include comparison with daily gauged (point) rainfalls and with gridded ERA-interim. We consider daily gauged point rainfalls as these represent the base data on which the AWAP estimates are derived, though as discussed in the paper we would expect there to be differences between point and gridded estimates as the latter data set accounts for some spatial averaging. Our rationale for including ERA-Interim is that it is has been found to be the best performing data set compared to other reanalysis rainfall products in the Australian region (Peña-Arancibia et al., 2013), and it shares similarity with BARRA in that rainfall observations are not assimilated. Importantly, ERA-Interim also provides the boundary condition for BARRA-R simulations and this allows the incremental value of BARRA-R to be assessed. Our focus with these comparisons is solely on the efficacy of the BARRA-R rainfall product, we do not set out to instruct users on the selection of wide-ranging precipitation products that are available across Australia.

Lastly, one additional reason for considering the performance of BARRA-R at a daily time step is that it has the potential to provide more accurate – or at least a credible alternative – estimate of daily rainfalls in regions which are sparsely gauged. There are large areas of Australia with very sparse gauging (see left-hand panel in the figure below), and it may be useful to supplement estimates of rainfalls in these regions with reanalysis products that do not rely on rainfall observations for assimilation. Without observations to inform surface fitting, it is evident that AWAP can yield erroneous results (see right hand panel in the figure below). One potential advantage of BARRA thus lies in its ability to complement the existing datasets in regions where gauged observations are sparse (especially in the semi-arid regions). In this regard, we note that only AWAP grid points that contain rainfall gauges are selected for comparisons in this study.

In the following sections we provide a detailed response to all the remarks raised by the referees. We had posted a short comment earlier to clarify the general concern raised by R1, and we now proceed to address all the comments by listing the reviewers' comments (RC, in blue), our corresponding reply (AR, in black), and proposed modifications in italics.

[Figure]

**Figure showing location of all daily rainfall stations in Australia (left panel) and AWAP estimates that erroneously indicate zero and less than 10mm total rainfalls over a twenty year period (right panel). The left panel is sourced from the Australian Bureau of Meteorology** (http://www.bom.gov.au/climate/data).

**Response to Referee #1**

**Response to general comment**

[RC] The manuscript "An evaluation of daily precipitation from atmospheric reanalyses over Australia" aims at comparing the new reanalysis precipitation dataset BARRA with ERA-Interim over Australia using in-situ rainfall data (point-to-grid analysis) and AWAP dataset (grid-to-grid analysis) as benchmarks. I do believe that the paper reads very well, it is properly structured and addresses a relevant topic of uttermost importance. The authors showed that the new dataset BARRA tends to outperform in most of the case ERA-Interim, while provides lower performances when compared to the AWAP dataset. In my opinion, I found the comparison of BARRA with only 1 reanalysis dataset not enough to justify a possible publication in HESS.

[AR] We thank Referee #1 for acknowledging the relevance and importance of the topic discussed.

**Responses to specific comments**

1.1 The purpose of the current study is to document the performance of the BARRA dataset at a daily scale and to provide a comparative analysis of its strengths and limitations relative to other available datasets. However, only ERA-Interim is used as a comparison. Why did the authors decide to use only 1 dataset for comparison? Why the choice of using another reanalysis dataset (ERA-Interim) and not other based purely on satellite product (e.g. PERSIANN) or corrected satellite (e.g. PERSIANNCDR)? I believe the manuscript (and the comparison) will benefit with the inclusion of additional recent and well-known datasets (e.g. CHIRPS, MSWEPv2.1, SM2RAIN ASCAT, CMORPH-CRT), or other reanalysis datasets (e.g. JRA-55, NCEP-CFSR, PFD, or WFEDEI GPCC) for comparison. Obviously, I am not suggesting to include several datasets in this analysis, but the comparison with 3 or 4 more datasets will definitely strengthen the impact of this research and manuscript.

[AR] As mentioned above, our aim is to evaluate the performance of BARRA dataset rather than provide instructive comments on use of wide-ranging precipitation products. We will provide additional clarification on these points in a revised version of the Introduction. Further, we propose to change the title of the paper to "*An evaluation of daily precipitation from a regional atmospheric reanalysis over Australia*" to better reflect the focus of our paper.

1.2 The authors first mentioned that "The accuracy at a daily scale provides us with an important benchmark as it is applicable to many hydrological applications and also forms the basis for further examination at finer timescales". However, the author then contradicted themselves concluding that "The core attraction of the BARRA dataset is the availability of sub-daily precipitation estimates. Such information is not available in the AWAP data set, and the spatial resolution of the estimates is higher than the currently available global 20 reanalysis and satellite datasets". In fact, in hydrological application at large scale (which is the case for the BARRA dataset due to a spatial resolution of 36km) daily time scale is most used temporal resolution. For this reason, as end-user, I would select the AWAP dataset as input for a large scale model as the resolution is higher and more appropriate to represent complex topographies. Beside the scientific interest in comparing different precipitation datasets, why someone should use BARRA if AWAP is already providing excellent performances at higher spatial resolution?

[AR] The choice of daily scale and its utility is addressed in the general response above. We have revised the introduction section of the manuscript (Page 4 Lines21-25), as follows:

*Despite the availability of BARRA precipitation as hourly values, we select a daily timescale to match the temporal resolution of the best available gridded reference dataset (AWAP). In addition, the accuracy at a daily scale serves as a desirable first step towards further examination at finer timescales because any reliance on sub-daily estimates necessarily depends on its ability to correctly represent daily precipitation. The additional value of evaluation at a daily scale is that it assesses the potential of BARRA to provide estimates of daily rainfall in the sparsely gauged regions across Australia.*

1.3 Results and discussions of grid-to-grid analysis are very brief and conclusions are somehow similar to the point-to-grid analysis in which BARRA gives better results than ERA: What is the additional value of including such analysis? It would be better to include more dataset for comparison (see the first point) and run only point-to-grid analysis.

[AR] We do agree that the results of grid-to-grid analysis are brief. However, the performance based on frequency metrics (wet day frequency, dry-wet transition probability, and wet-wet transition probability) are similar for both grid-to-grid and point-to-grid analysis and leads to similar conclusions. It is thus appropriate to present the similar results in a brief manner. However, KGE' and its components at grid-to-grid evaluation shows that BARRA exhibits superior performance at grid scale (5 km) which is not obvious from point-to-grid analysis. Given the spatial averaging that is implicit in grid-based products it is not expected that similar results would be obtained with point-to-grid comparisons, or with grid-to-grid comparisons at different resolutions (AWAP is 5 km, BARRA-R is 12km, and ERA-Interim is 80km). This information is reflected in the revised introduction (Page 4 Lines11-16) and methods section (Page 6 Lines 13-21):

*The evaluation is performed against gauged measurements from two data sources: one is based on gauged point observations, and the other is a high-resolution gridded AWAP dataset derived from interpolating gauge measurements, which is widely accepted as being the best synthesis of gauged observations. There are inherent differences in gridded and point rainfall estimates due to the spatial averaging of point observations across each grid cell area. Since BARRA provides direct estimates of area-average rainfall, it is useful to compare these estimates with the best available point and areal precipitation estimates.*

*…*

*As the gridded datasets represent an areal average, it may be expected that there are differences between point and gridded estimates as the latter account for some spatial averaging. While the gauged data and AWAP rainfall estimates represent the best available reference datasets based on measured data, both are imperfect representations of areal rainfalls. The AWAP estimates contain inaccuracies due to the interpolation method, and the point estimates provide only a coarse estimate of rainfall over a grid cell area. The ability of these point and gridded reference data sets to represent actual areal rainfalls is heavily dependent on the gauging density and local orography, and these factors influence*

*the accuracy of the reference data sets to different degrees across Australia. Accordingly, we compare the BARRA and ERA-Interim estimates to both point gauged and AWAP areal data, where the evaluation of both offer different insights to the quality of model estimates.*

1.4 How the different spatial resolution of the reanalysis dataset and interpolation method of the in-situ gauges can affect the results of this analysis?

[AR] The effect of spatial resolution is clearly observed in the comparison of quantiles: coarser datasets are not well suited to representing large rainfall due to averaging over larger area. In addition, the choice of interpolation could also affect the analysis of large rainfalls. In this study, nearest neighbour interpolation is used which preserves the magnitude of precipitation (averaged) over the grid, however, bilinear interpolation is likely to smooth the precipitation field resulting in higher bias for larger rainfall (Accadia et al., 2003).

We have added the reasoning for our choice of interpolation methods in section 3.1 (Page6 Lines 22-26):

*Gridded rainfalls are compared to point rainfalls using a nearest neighbour approach. The choice of interpolation scheme is especially important when comparing datasets of different spatial resolutions; bilinear interpolation is likely to smooth the precipitation field resulting in higher bias for larger rainfall (Accadia et al., 2003), whereas nearest neighbour method preserves the magnitude of the precipitation over the grid.*

1.5 The methodology is quite straightforward and based on existing approaching for comparing distributed precipitation dataset. Besides the comparison of different datasets over Australia using different performance measures which one is the main research innovation of this paper?

[AR] We agree that the methodology in this study is straightforward and based on existing approaches for evaluation of precipitation dataset. However, the precipitation dataset used in this study is new and is the only regional reanalysis product developed for the Australasian region. The results of this study are useful for the potential users of the dataset for hydrometeorological studies in the Australasian region.

1.6 From Figure 2.d it is difficult to assess where BARRA is performing (on average) better than ERA-Interim. From my point of view, ERA-Interim shows overall higher KGE values than BARRA (blue points). I suggest the authors to estimate the average (and standard deviation) of the values in figure 2.d to see which dataset provides higher KGE.

[AR] The summary of difference in KGE has now been added in the revised plot (Figure 2d)

1.7 Lines 18-20 page 7 "Despite having a slightly lower correlation compared to ERAInterim, the variability of the rainfall is better captured by the BARRA dataset" I do not agree with the authors. From figure 3 it looks that ERA tends to outperform BARRA in almost all the considered performance measures. Also, how the authors can say that rainfall is better captured by BARRA dataset if an aggregated index (KGE) is used?

[AR] We checked the plot and the corresponding statement. The best value for variability ratio is 1 and the variability ratio in BARRA is closer to 1 compared to ERA-Interim in overall as well as seasonal analyses. We have added the lines representing best values in KGE' as well as its components to add interpretation of the plot (Figure 3).

1.8 Line 14, page 8 "The spatial pattern, however, is similar for all datasets." Not really. It looks to me that spatial pattern is different from figure 4. Are the authors referring only to the spatial pattern of BARRA and ERA-Interim?

[AR] Based on figure 4, there is slight difference in magnitude of the frequencies across datasets. We agree that BARRA and ERA-Interim are very close to each other. But, in terms of spatial pattern, all three datasets behave similarly as observed in spatial plot shown below.

[Figure]

**Figure showing frequency of wet days for AWAP, BARRA, and ERA-Interim at gauge locations used in this study.**

**Response to Referee #2 Korbinian Breinl**

**Response to general comment**

[RC] The present article deals with the evaluation of a new reanalysis dataset called BARRA with other gridded datasets (ERA-Interim and AWAP) and rain gauge observations, for Australia. This is a well written paper, which easy to read. At this stage I think however that some improvements are needed for the publication in HESS.

[AR] Thank you for your comments.

**Responses to specific comments**

2.1 I wonder if the results of the mean precipitation (Figure 1 and related text in the results section) could be presented in a different way – besides the four maps that are useful for sure – to better capture the changes in spatial variability. Maybe the authors have a good idea for a plot.

[AR] The four maps in Figure 1 visually demonstrate the spatial variability in precipitation across datasets. In addition, we have now computed a correlation measure between BARRA-R and AWAP, and similarly between ERA-Interim and AWAP. We have added the plot in the supplement (Figure S1).

Also, I wonder if cutting off the over-sea precipitation for BARRA and ERA-Interim would help to better read the map, meaning taking the over-land precipitation as the lowest common denominator.

[AR] Our original logic for retaining this was two-fold: to visualize spatial richness in the datasets, and to show that BARRA extends across ocean surrounding Australia which is not the case for gauged measurements and AWAP. However, we are happy to mask the information shown on these plots so they cover the same physical domain. The subplots in the revised plot (Figure 1) now represents precipitation over land for all the datasets.

2.2 I think it would be good to look into spatial correlations. As far as I can see you have not looked into them, although they are relevant. Is there a particular reason for not taking them into account?

[AR] The correlation coefficient was computed for each station and their spatial distribution was visualised. It was not shown in the manuscript as the spatial distribution of correlation coefficient was very close to spatial distribution of KGE'. We have not looked further into spatial correlations, though as responded in 2.1, the added correlation plot (Figure S1) will demonstrate the correlation structure across space.

2.3 I do not fully understand the purpose of the BARRA dataset, at least not in the context of the article, which is focusing on daily values. Considering that the AWAP dataset is superior to BARRA (superior at daily timescale), why would I use BARRA (when sub-daily is not the topic)?

[AR] We address the choice of daily scale and its utility in the general response above. In addition, we have made modifications to the introduction section to clarify the choice of daily timescale to our readers (Page 4 Lines 21-25):

*Despite the availability of BARRA precipitation as hourly values, we select a daily timescale to match the temporal resolution of the best available gridded reference dataset (AWAP). In addition, the accuracy at a daily scale serves as a desirable first step towards further examination at finer timescales because any reliance on sub-daily estimates necessarily depends on its ability to correctly represent daily precipitation. The additional value of evaluation at a daily scale is that it assesses the potential of BARRA to provide estimates of daily rainfall in the sparsely gauged regions across Australia.*

 What exactly is the added value of comparing the gridded data among each other, without the point rainfall? I would like to try to understand the motivation behind it, isn't the comparison with the point rainfall sufficient enough?

[AR] As noted earlier, BARRA represent area-average rainfall: while gauged data represents the best estimate of rainfall at a point location, AWAP serves as the best available estimate of areal rainfall. We think it is useful to undertake both grid-to-point and grid-to-grid comparisons as both types of estimates are likely to contain biases due to differences in spatial averaging, and the levels of measurement support also differs between the different products. The two analyses thus offer different insights to the quality of the model estimates. It is worth noting that in terms of performance, KGE' and its components at grid-to-grid evaluation shows that BARRA has superior performance at grid scale (5 km) which is not obvious from point-to-grid analysis.

This is further clarified in the revised introduction (Page 4 Lines11-16) and methods section (Page 6 Lines 13-21):

*The evaluation is performed against gauged measurements from two data sources: one is based on gauged point observations, and the other is a high-resolution gridded AWAP dataset derived from interpolating gauge measurements, which is widely accepted as being the best synthesis of gauged observations. There are inherent differences in gridded and point rainfall estimates due to the spatial averaging of point observations across each grid cell area. Since BARRA provides direct estimates of area-average rainfall, it is useful to compare these estimates with the best available point and areal precipitation estimates.*

*...*

*As the gridded datasets represent an areal average, it may be expected that there are differences between point and gridded estimates as the latter account for some spatial averaging. While the gauged data and AWAP rainfall estimates represent the best available reference datasets based on measured data, both are imperfect representations of areal rainfalls. The AWAP estimates contain inaccuracies due to the interpolation method, and the point estimates provide only a coarse estimate of rainfall over a grid cell area. The ability of these point and gridded reference data sets to represent actual areal rainfalls is heavily dependent on the gauging density and local orography, and these factors influence the accuracy of the reference data sets to different degrees across Australia. Accordingly, we compare the BARRA and ERA-Interim estimates to both point gauged and AWAP areal data, where the evaluation of both offer different insights to the quality of model estimates.*

2.5 In that context (point 4), it would be good to see what has already been done in terms of evaluations of gridded rainfall vs. point rainfall (and also grid rainfall to other grid rainfall), as conducted in this study, means I would like to see a more comprehensive literature review. A brief review of such evaluations helps the reader to better understand the implications of the present study. For example, what are the pros and cons of ERA-Interim according to other studies, and what has been concluded in this article? Again, I miss a bit the ability to generalize from the results from this article.

[AR] We have added further details of relevant literature that contain grid-to-grid and point-to-grid evaluations. We have included reference to the studies that document the performance of ERA-Interim (in global and Australian region) and the Unified Model on which the BARRA-R estimates are derived.

The revised text in the introduction section (Page 2 Lines 16-31, Page 3 Lines 1-9) now reads:

*Global reanalysis datasets have been evaluated for various applications at global and regional scales. Of the global dataset exclusively based on reanalysis, ERA-Interim is generally considered to provide better performance compared to other reanalysis datasets (Beck et al., 2017, 2019). ERA-Interim has been found to reproduce the climatology of global monsoon precipitation (Lin et al., 2014) and in general it demonstrates high temporal and spatial correlations with interpolated observations (Donat et al., 2014). In a recent global evaluation of gridded precipitation datasets using gauge observations by Beck et al (2017), the ERA-Interim and JRA-55 reanalysis datasets were found to reproduce long-*

*term trends and temporal correlation more reliably than achieved by satellite datasets. In the Australian continent, ERA-Interim reproduced the observed spatial patterns of long-term rainfall along with other climatic variables and showed an overall better performance compared to NCEP-NCAR (National Centers for Environmental Prediction/National Center for Atmospheric Research) reanalysis (Fu et al., 2016). An evaluation by Peña-Arancibia et al. (2013) of reanalysis datasets, satellite products, and an ensemble of these datasets in Australian and Asian regions showed that the ERA-Interim performed better than other individual datasets across a range of metrics for the Australian region.*

*The available global reanalysis datasets such as NCEP-CFSR, ERA-Interim, and JRA-55 cover the Australian region, but their horizontal resolutions are relatively coarse (≥ 80 km) and unsuitable for fine-scale application in hydro-meteorological analysis. The resolution of the global reanalysis can be enhanced by downscaling approaches such as dynamic downscaling (Soares et al., 2012) or statistical analysis using high-resolution surface observations (Vidal et al., 2010). Alternatively, the application of a regional model-based data assimilation can provide a better representation of local climate features and extreme events (Bollmeyer et al., 2015). The regional reanalysis, unlike global reanalysis, allows the integration of abundant local surface observations at a finer scale (Bollmeyer et al., 2015; Isotta et al., 2015; Jakob et al., 2017). The studies have also been conducted to evaluate the additional benefit of high-resolution datasets obtained from regional reanalysis. Jermey and Renshaw (2016) found that the overall performance of 12km regional reanalysis was significantly better than 80km ERA-Interim especially for high thresholds of rainfall. Similarly, the performance of a regional analysis against ERA-Interim at grid scale showed an improvement in representing high-threshold events (Isotta et al., 2015; Jermey and Renshaw, 2016; Roberts and Lean, 2008) and convective events (Roberts and Lean, 2008). In general, regional reanalyses based on boundary conditions from ERA-Interim is shown to improve the performance over ERA-Interim albeit these studies are largely focused in Europe.*

2.6 I would appreciate a final overview that summarizes the results, ideally in the format of a table. The table could contain for example (i) general information for each dataset (time period, spatial resolution, URL to obtain data), (ii) metrics for evaluation, (iii) performance of each metric etc. The table would make it much easier to get a good overview of the results. And, even better, another column could add some information on each dataset or similar datasets (similar in terms of how they were processed) and some results from other studies if applicable (see comment above).

[AR] We have added a table with information on datasets and another one with summary of performance in the revised manuscript. The added tables are:

*Table 1 Overview of gridded precipitation datasets used in this study*

| Name | Details | Data Source | Spatial coverage and resolution | Temporal coverage and resolution | Reference |
|---|---|---|---|---|---|
| *BARRA-R* | *Bureau of Meteorology Atmospheric high-resolution Regional Reanalysis for Australia* (http://www.bom.gov.au/research/projects/reanalysis/) | *Regional reanalysis* | *Australasian region (65 to 196.9° E, -65 to 19.4° N), ~12km* | *1990-February 2019, Hourly* | *(Su et al., 2019)* |
| *ERA-Interim* | *European Centre for Medium-range Weather Forecasts ReAnalysis Interim* | *Global reanalysis* | | *1979-present,* | *(Dee et al., 2011)* |

| | *(https://www.ecmwf.int/en/forecasts/datasets /reanalysis-datasets/era-interim)* | | *Global, ~80km* | *3-hourly* | |
| --- | --- | --- | --- | --- | --- |
| *AWAP* | *Australian Water Availability Project* *(http://www.csiro.au/awap/)* | *Gauge interpola ted* | *Australia land area, 5km* | *1900 – present, Daily* | *(Jones et al., 2009)* |

*Table 2 Summary of performance of BARRA precipitation*

| Metric | Point-to-grid | Grid-to-grid |
| --- | --- | --- |
| *KGE' and its components* | *Performance varies spatially, showing better scores in temperate regions than in the tropical and arid region* *Variability is well captured* | *Exhibits less bias in total rainfall than ERA-Interim* *Overestimates variability in the tropics only* |
| *Quantiles* | *High quantiles (90, 95 and 99%) closely represent high rainfalls and are unbiased.* | *Similar pattern as for point-to-grid* *Better representation of higher quantiles than ERA-Interim* |
| *Wet day frequency and transition probabilities* | *Frequency of wet days and wet-wet transition probabilities are over-estimated* *Dry-wet transition probabilities are well reproduced* | *Similar pattern as for point-to-grid with improved correlation and reduced bias.* |
| *Categorical metrics* | *Both POD and FAR increase with threshold* *Frequency bias is on average higher for light/moderate than low and heavy rainfalls.* | *Improvement along all metrics with respect to ERA-Interim* *Light rainfalls are over-estimated* *Large rainfalls are poorly captured yet are better than ERA-Interim* |

2.7 Reviewer #1 asked for innovation. This is not my main concern as long as I can draw general conclusions from this paper as a reader for my own studies (also outside Australia), but at this stage I miss it a bit. The article tries to explain the results to a certain degree, but I think the discussion should be more detailed, and pros and cons of each dataset should be better explained considering how the datasets were generated, considering seasons, resolution etc. I find statements such as "AWAP estimates of point rainfall are higher than the gauged observations" or "ERA-Interim generally performs better in the central arid region, whereas BARRA exhibits better scores in the temperate region", but I miss good explanations why. The table overview I addressed may help to make this generalization easier. Also, I must admit, looking into more reanalysis data (ideally popular datasets) as suggested by Reviewer #1 may make the paper stronger – at least one more prominent dataset maybe?

[AR] Thank you for acknowledging the relevance of the study. We agree that significance of this paper is enhanced by its ability to provide general conclusions useful for other studies, different data sets, or study areas. In that regard, we have revisited the datasets and discussion sections and explained the physical basis of the results further in order to aid generalization of the research beyond current study area As mentioned in the general response, our focus is on evaluating the new BARRA-R reanalysis dataset rather than providing instruction to users on selection of available precipitation datasets.

Adoption of a changed title (as discussed in response to 1.1 above) should better reflect the intent of the paper.

We have added a discussion of BARRA (Page 5 Line 24 – Page 6 Line 2):

*The model includes a comprehensive set of parametrisations, including a modified boundary layer scheme, mixed phase cloud microphysics, a mass flux convection scheme, and a radiation scheme. The model parametrisation in BARRA mainly is inherited from the UKMO Global Atmosphere (GA) 6.0 configurations as described in Walters et al. (2017). Surface and satellite rainfall observations are not assimilated, and the precipitation fields are determined using the assimilated large-scale variables and the physical parameterisation of the model. At 12 km horizontal resolution, BARRA requires the convection scheme to model sub-grid scale convection using an ensemble of cumulus clouds as a single entraining-detraining plume (Clark et al., 2016). The scheme prevents unstable growth of cloud structures on the grid and explicit vertical circulations and can only predict an area-average rainfall instead of a spectrum of rainfall rates.*

We have also added the following in the Results section:

*(Page 8 Lines 21-26) ERA-Interim generally performs better in the central arid region, whereas BARRA exhibits better scores in the temperate region. The central arid region experiences very few rainy days which is likely to be missed spatially by a sparse network of gauges. This could result in poor correlation metrics as it is sensitive to outliers, and subsequently yields a poor KGE' score. BARRA precipitation is more erroneous in the tropics than in temperate regions which may reflect the limitations of the convection parameterisation adopted in BARRA. This spatially varying performance of BARRA is further discussed in section 5.5).*

*(Page 9 Lines 2-5) The AWAP estimates of point rainfall are higher than the gauged observations, and a similar degree of overestimation is exhibited by the BARRA dataset. The Barnes interpolation scheme used in AWAP slightly inflates the spatial coverage of light intensity rainfall (Jones et al., 2009; King et al., 2013). In addition, the weight function assigned to gauged point rainfalls may result in wetter bias in the densely gauged (coastal) regions compared to other regions.*

*(Page 9 Lines 8-10) The performance during winter is better than in summer for both reanalysis datasets. This is likely due to the ability of NWP models to accurately simulate synoptic systems which represent the majority of wintertime rainfall.*

*(Page 10 Lines 2-4) The model estimates of transition probability $p_{11}$ are greater than $p_{01}$. Due to the tendency of NWP models to yield more frequent persistent light rainfalls, a high wet-wet frequency is observed.*

We have added the following in the Discussion section.

*(Page 13 Lines 28-30) The spread in transition probabilities $p_{01}$ is estimated well by all gridded datasets, whereas the $p_{11}$ estimates are overestimated and more varied. As mentioned above, this difference is due to the tendency of NWP models to over-estimate the persistence of light rainfalls (Kendon et al., 2012).*

*(Page 14 Line25 – Page 15 Line 1) The possible explanation for this may be the difference in the climatic systems driving the precipitation in those regions and the scheme used to generate precipitation (Su et al., 2019). Convective precipitation is dominant in tropical regions and the parameterisation scheme for sub-grid convection adopted in BARRA is limited in terms of resolving such precipitation. Therefore, a higher accuracy of BARRA can be expected at high latitudes where synoptic rainfall dominates than in low latitudes where convective rainfall is dominant (Ebert et al., 2007; Su et al., 2019). The limitation of NWP models to represent convective precipitation accurately has also been reported by Ebert et al. (2007) and de Leeuw et al. (2015).*

2.8 Figure 6. I would add x-axis labels to the upper plots

[AR] Agreed, we have added the x-axis labels in the revised plot.

2.9 Optional: The code (R, MATLAB etc.) to analyse the rainfall data could be published with the paper. I always encourage to do so, but this is up to the authors of course.

[AR] We will identify the codes that we consider most relevant for general use. We will upload the codes as a public GitHub repository and add the corresponding link in the manuscript under the sub-section "Data Availability".

Other Changes not mentioned above

1. The Table 1 in the first manuscript which summarizes the performance metrics used in the study is removed from the main text and added as a supplement.

[revised manuscript text omitted]

---

## Author Response (AR2)

**Response to reviews on the manuscript hess-2018-607 "An evaluation of daily precipitation from atmospheric reanalyses over Australia" by Suwash Chandra Acharya et al.**

We agree with the Editor's suggestion to publish the codes along with the paper.

5     We have updated the supplement to include the codes used for the analysis and relevant for general use. We have added a section "Code availability" in Page 16 Line 10.

[revised manuscript text omitted]